# Activity disruption causes degeneration of entorhinal neurons in a mouse model of Alzheimer's circuit dysfunction

**Rong Zhao[1†], Stacy D Grunke[1†], Caleb A Wood[1†], Gabriella A Perez[1], Melissa Comstock[1], Ming-Hua Li[1], Anand K Singh[1], Kyung-Won Park[1], Joanna L Jankowsky[1,2]***

[1]Department of Neuroscience, Huffington Center on Aging, Baylor College of Medicine, Houston, United States; [2]Departments of Neurology, Neurosurgery, and Molecular and Cellular Biology, Huffington Center on Aging, Baylor College of Medicine, Houston, United States

**Abstract** Neurodegenerative diseases are characterized by selective vulnerability of distinct cell populations; however, the cause for this specificity remains elusive. Here, we show that entorhinal cortex layer 2 (EC2) neurons are unusually vulnerable to prolonged neuronal inactivity compared with neighboring regions of the temporal lobe, and that reelin + stellate cells connecting EC with the hippocampus are preferentially susceptible within the EC2 population. We demonstrate that neuronal death after silencing can be elicited through multiple independent means of activity inhibition, and that preventing synaptic release, either alone or in combination with electrical shunting, is sufficient to elicit silencing-induced degeneration. Finally, we discovered that degeneration following synaptic silencing is governed by competition between active and inactive cells, which is a circuit refinement process traditionally thought to end early in postnatal life. Our data suggests that the developmental window for wholesale circuit plasticity may extend into adulthood for specific brain regions. We speculate that this sustained potential for remodeling by entorhinal neurons may support lifelong memory but renders them vulnerable to prolonged activity changes in disease.

**\*For correspondence:**
jankowsk@bcm.edu

[†]These authors contributed equally to this work

**Competing interest:** The authors declare that no competing interests exist.

## Editor's evaluation

This is a fundamental study that demonstrates that ongoing neuronal activity plays a key role in the vulnerability of specific neuronal cell types in layer 2 of the entorhinal cortex that communicates with the hippocampus. The authors provide compelling evidence that chronic silencing of excitatory neurons in the entorhinal cortex leads to their degeneration. Reelin-positive stellate neurons were the most vulnerable to silencing. The authors propose that developmental mechanisms associated with activity-dependent circuit refinement could be aberrantly reactivated in the context of Alzheimer's disease.

## Introduction

Each neurodegenerative disease is characterized by selective loss of a specific neuronal population. Current research efforts have focused heavily on the molecular features of these neurons that render them susceptible to disease pathology. Modern transcriptomic techniques have revealed distinct gene signatures of vulnerable neuronal populations, but this approach takes a static view of circuits that remain active throughout life. Neuronal activity plays a key role in circuit refinement during development up to and including cell death, raising the possibility that developmental mechanisms could

**eLife digest** Neurodegenerative conditions cause irreversible damage to the brain and have a devastating impact on quality of life. However, these diseases start gradually, meaning that the entire brain is not affected at once. For example, the initial signs of Alzheimer's disease appear only in specific areas.

One of the first brain regions to degenerate in Alzheimer's is the entorhinal cortex. In healthy individuals, entorhinal neurons send electrical signals to the hippocampus, a part of the brain important for memory and learning. During Alzheimer's, hippocampal neurons also die off, leading to 'shrinkage' of this brain region and, ultimately, the memory problems that are a hallmark of the disease.

Many neurons in the developing brain require electrical input from other cells to survive – in other words, if they do not belong to an 'active circuit', they are eliminated. This is crucial for the connection between the entorhinal cortex and the hippocampus, where the circuit's development and maintenance require carefully controlled electrical activity. Abnormal electrical activity is also an early sign of diseases like Alzheimer's, but how this relates to degeneration is still poorly understood.

By investigating these questions, Zhao, Grunke, Wood et al. uncovered a potential relationship between electrical activity and degeneration in the adult brain, long after the circuit between the hippocampus and the entorhinal cortex had matured.

Mice were genetically engineered so that their entorhinal cortex would carry a protein designed to silence electrical signaling. The communication between the entorhinal cortex and the hippocampus could therefore be shut off by activating the protein with an injected drug. Remarkably, within just a few days of silencing, cells from the entorhinal cortex started to die off.

Zhao, Grunke, Wood et al. went on to show that different silencing methods yielded the same results – in other words, the degeneration of cells from the entorhinal cortex was not linked to a particular method. This vulnerability to electrical inactivity was also unique to the entorhinal cortex: when neighboring parts of the brain were silenced, the nerve cells in these areas did not die as readily. Interestingly, in one of their experiments, Zhao, Grunke, Wood et al. found that electrical activity of neighboring nerve cells participated in killing the silenced neurons, suggesting that nerve cells in these brain areas might compete to survive.

Overall, this work highlights a direct link between electrical activity and nerve cell degeneration in a part of the brain severely affected by Alzheimer's. In the future, Zhao, Grunke, Wood et al. hope that these results will pave the way to a better understanding of the biological mechanisms underpinning such neurodegenerative diseases.

be aberrantly reactivated in the context of disease. During CNS development, some regions eliminate up to 50% of immature neurons through activity-dependent programmed cell death (*Murase et al., 2011*; *Wong and Marín, 2019*). Still more complex and varied processes drive axonal refinement in development (*Riccomagno and Kolodkin, 2015*; *Neukomm and Freeman, 2014*). Despite being one of the strongest factors in developmental circuit refinement, neuronal activity has not been well studied as a potential contributor to neurodegeneration (*Gonzalez-Rodriguez et al., 2020*; *Wang and Holtzman, 2020*).

Changes in neuronal activity are an early hallmark of several degenerative disorders, and it is increasingly clear that network dysfunction can drive pathogenesis (*Palop and Mucke, 2016*; *Harris et al., 2020*; *Wu et al., 2016*; *Yamamoto et al., 2015*; *Yamada et al., 2014*; *Cirrito et al., 2005*; *Yuan and Grutzendler, 2016*; *Rodriguez et al., 2020*). In Parkinson's disease, vulnerability of dopaminergic neurons has been attributed to the high-energy demand of maintaining persistent low-frequency firing across a vast axonal arbor (*Gonzalez-Rodriguez et al., 2020*). The normal function of nigral neurons may render them preferentially vulnerable to degeneration. In Alzheimer's disease (AD), amyloid formation can lead to hypersynchronous activity, which is thought to accelerate cognitive decline (*Cirrito et al., 2005*; *Vossel et al., 2016*; *Palop and Mucke, 2016*). Increased neuronal activity promotes further amyloid release, creating a vicious cycle (*Cirrito et al., 2005*; *Styr and Slutsky, 2018*; *Frere and Slutsky, 2018*). In both of these settings, neuronal activity promotes pathogenesis. We were therefore surprised to uncover evidence that neuronal inactivity contributes to neurodegeneration in a model of AD. While modeling the circuit-level consequences of neurodegeneration in the

entorhinal cortex (EC), we uncovered evidence that this area is highly susceptible to activity disruption well into adulthood. Using multiple chemogenetic systems, we show that electrically silencing entorhinal neurons elicits a pattern of degeneration beginning with axon damage, followed by neuroinflammation and overt cell death. Inhibiting synaptic vesicle release without altering electrical activity is sufficient to drive entorhinal death, but triggers a distinct pattern of axon degeneration. Finally, we found that axonal degeneration following synaptic inhibition is governed by classic interneuronal competition and can be prevented by broadly silencing all entorhinal neurons. Together, our studies indicate that neuronal activity plays an ongoing role in the survival of EC neurons and raise the possibility that perturbations in activity may partner with molecular pathology to drive degeneration of this region.

## Results

### Unexpected loss of EC2 neurons following chloride-based neuronal silencing

In previous work, we described a chemogenetic model for studying neuronal dysfunction in the adult EC (*Zhao et al., 2016*). This model used an engineered chloride channel to allow ligand-activated disruption of action potential initiation in a subset of EC layer 2 neurons (EC2) (*Lynagh and Lynch, 2010*, *Lynagh and Lynch, 2012*). The modified channel (GlyCl) was expressed in EC2 neurons using a tetracycline-transactivator driver line that was selective for, but not exclusive to, EC2 cells (Nop-tTA; TRE-GlyCl-YFP) (*Yasuda and Mayford, 2006*; *Yetman et al., 2015*). The GlyCl channel is activated by the ligand ivermectin (IVM) and was co-expressed with yellow fluorescent protein (YFP) to identify the silenced cells. Bath application of 100 nM IVM to acute brain slices suppressed neuronal firing by up to 90% in YFP+ entorhinal neurons from GlyCl transgenic mice, but had no effect on EC neurons from control animals (*Figure 1A*). We previously demonstrated that systemic administration of IVM decreased neuronal firing in vivo for several hours after intraperitoneal injection and that the drug half-life of >12 hr likely maintained this effect for a day or more (*Zhao et al., 2016*).

Following the initial description of this new chemogenetic silencing system, we discovered an unexpected change in the labeling of GlyCl + EC neurons over the week following IVM injection. Vehicle-injected animals displayed clear YFP labeling of neuronal cell bodies in EC2 and their axonal projections into the dentate gyrus (DG) (*Figure 1B*). In contrast, animals injected with a single dose of IVM (5 mg/kg, i.p.) showed diminished DG axonal labeling within 1–2 days post injection (dpi) and labeling was nearly undetectable by 4 dpi. Alongside this change in the DG, the number of labeled cell bodies in EC2 was significantly decreased by 4 dpi and reduced by 50% at 7 dpi (*Figure 1B*).

The spatiotemporal pattern of diminished labeling suggested that the silenced cells were being eliminated from the circuit. To confirm that the silenced neurons had degenerated, we began looking for canonical markers of neuronal injury and cell death. First, we saw that many YFP-labeled EC2 axons had withdrawn from the DG into the cortex and developed pronounced swellings at their distal ends. These swellings were strongly reminiscent of retraction bulbs, which are typically observed after axonal injury or physical lesion (*Hill et al., 2016*; *Figure 1C*). These bulbs appeared within 1 day of IVM treatment, peaked at 2–3 dpi, and were gone over the same time course as the loss of YFP-labeled cells (*Figure 1C*). Retraction bulbs began while IVM was still active in the brain, suggesting that their emergence was the result of neuronal silencing and not due to rebound activity (*Zhao et al., 2016*) . Next, we found EC2 cells labeled with active caspase-3, indicative of apoptotic cell death, in the days immediately following IVM injection (*Figure 1B*). Caspase-labeled cells were sporadic, but only observed in EC2 of animals subjected to entorhinal silencing. Consistent with caspase activation, we observed that EC2 cells contained a caspase-cleaved actin fragment (fractin), again suggesting apoptotic cell death (*Schulz et al., 2009*; *Yang et al., 1998*; *Figure 1D*). Fractin labeling was robust, peaked between 4 and 7 days after IVM and then abated, presumably after the dead and dying cells had been cleared. Finally, we observed hypertrophic Iba1+ microglia in both the DG and the EC. Iba1 staining increased by 2 days after IVM, peaked by 4–7 dpi, and receded by 10–13 dpi (*Figure 1D*). Collectively, our data suggests that transient neuronal silencing of EC2 neurons initiated a cascade of events beginning with axonal retraction and ending in caspase-associated cell death with a strong but focal microglial response.

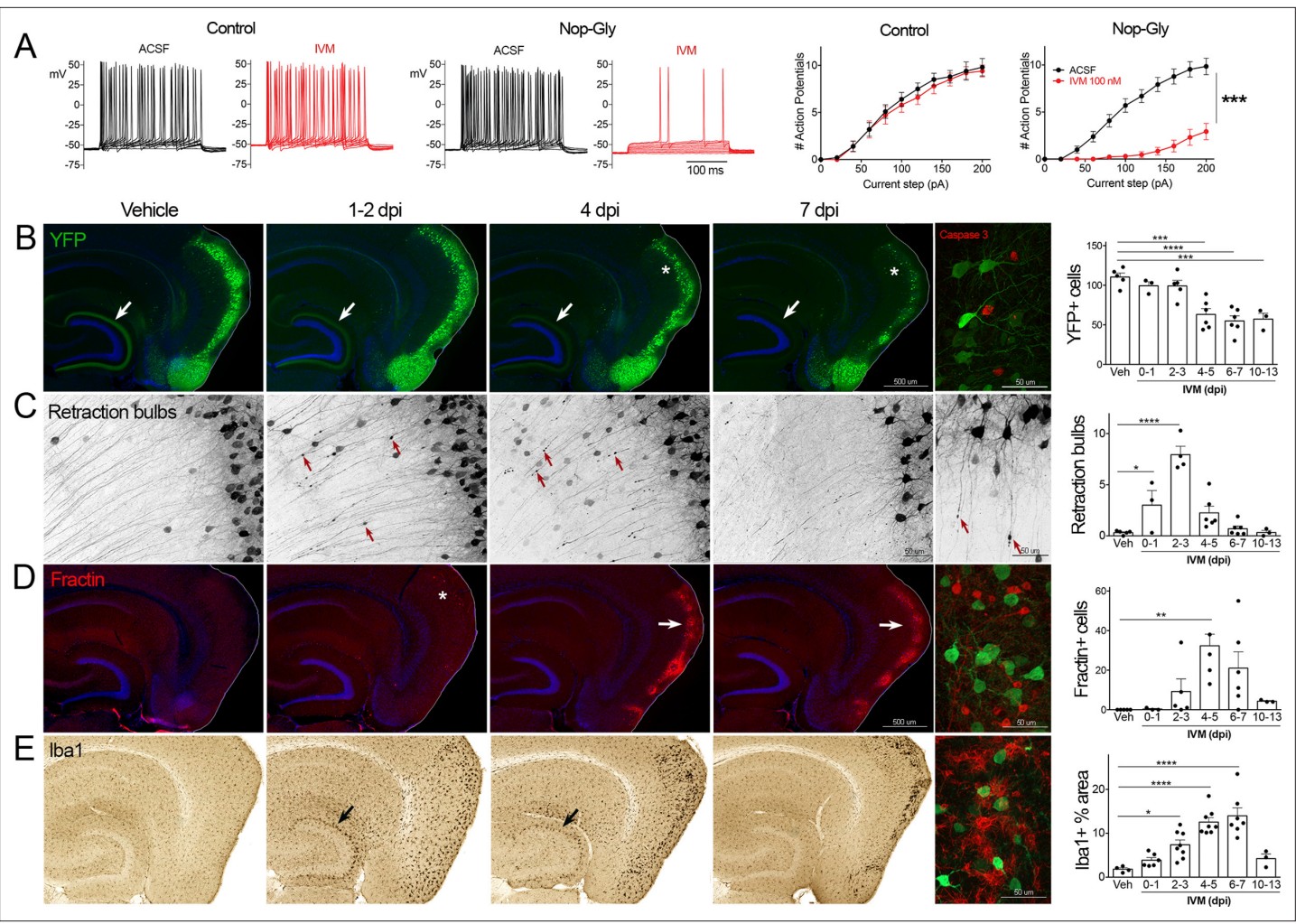

**Figure 1.** Transient electrical silencing with GlyCl causes cell death in entorhinal cortex (EC) neurons. (**A**) Example traces from current-clamp recordings in brain slices from control and GlyCl-expressing mice, with and without 100 nM ivermectin (IVM). Graphs on right show the average number of action potentials elicited at each current step. Data expressed as mean ± SEM, two-way repeated-measures ANOVA (p-value for interaction of treatment × current). n = 10 cells from 10 slices (five mice, control WT or single transgenic); 13 cells from 13 slices (six mice, Nop-tTA; TRE-GlyCl-YFP bigenic). (**B**) YFP immunostaining reveals that silencing EC neurons lead to progressive loss of dentate innervation (white arrow) followed by loss of YFP-labeled neuronal cell bodies in the EC (asterisk). Final image shows the presence of active caspase-3 immunostaining (red) among YFP+ cell bodies (green) at 3 days after IVM injection (dpi). (**C**) High-magnification images in EC show retraction bulbs on YFP-labeled axons withdrawing from the dentate gyrus (DG) (red arrows). Final image shows single YFP-labeled axons terminating in retraction bulbs at 2 dpi. (**D**) Fractin immunostaining marks the emergence of damaged/dying neurons in EC. Final image shows fractin+ cells (red) among YFP+ cells at 7 dpi. (**E**) Iba1 immunostaining reveals early microglial activation upon denervation of the DG (black arrow) followed by pronounced reactivity in the superficial EC. Final image shows Iba1 (red) and YFP (green). Graphs show average values per section for each animal ± SEM. n = 4–5 (veh), 3–6 (0–1 dpi), 4–8 (2–3 dpi), 4–8 (4–5 dpi), 6–7 (6–7 dpi), and 3 (10–13 dpi). One-way ANOVA with Dunnett's post hoc test, *p<0.05, **p<0.01, ***p<0.001, ****p<0.0001. Unnecessary cerebellar tissue, where present, was masked in Photoshop to match fields of view across panels. Associated with *Figure 1—figure supplements 1 and 2*.

The online version of this article includes the following source data and figure supplement(s) for figure 1:

**Source data 1.** Numerical data for *Figure 1*.

**Figure supplement 1.** Ivermectin (IVM) does not trigger neuronal loss in the absence of GlyCl.

**Figure supplement 1—source data 1.** Numerical data for *Figure 1—figure supplement 1*.

**Figure supplement 2.** Gradual loss of dentate gyrus (DG) labeling upon entorhinal cortex (EC) neuronal silencing with Kir.

**Figure supplement 2—source data 1.** Numerical data for *Figure 1—figure supplement 2*.

## EC2 cell death is observed with other means of electrical inactivation

The loss of EC2 neurons after GlyCl silencing was unexpected, and before investigating potential biological mechanisms, we wanted to rule out the possibility that we had simply generated a chemogenetic system for cell death. Although IVM had never been associated with neurodegeneration, we first confirmed that the drug by itself was not responsible for cell loss in EC. To test this, we introduced a nuclear-localized lacZ transgene into Nop-GlyCl animals that would generate two sets of mice: one with EC2 neurons co-expressing GlyCl plus lacZ, and a set of siblings that expressed only lacZ. We again found that EC2 cells were diminished in animals carrying both lacZ and GlyCl but detected no cell loss in mice expressing lacZ alone (*Figure 1—figure supplement 1*). This data demonstrates that GlyCl-based neuronal silencing and not IVM exposure alone was responsible for cell death in the Nop-GlyCl mice.

We next sought to determine whether EC2 neurons were selectively sensitive to chloride flux caused by the GlyCl system or whether instead they were more broadly vulnerable to electrical silencing. We chose a mutated Kir 2.1 channel to suppress firing by potassium shunting (*Xue et al., 2014*), and stereotaxically injected AAV carrying a tTA-dependent Kir2.1 into the EC of Nop-tTA single transgenic mice (*Figure 1—figure supplement 2A*). Viral expression peaked at approximately 7 dpi, when field recordings of input–output responses in the perforant path demonstrated a significant reduction in transmission between EC2 and DG (*Figure 1—figure supplement 2B*). Co-expressed YFP again allowed us to visualize the loss of axonal labeling from the DG over the following weeks that was qualitatively similar to, but slower than, what we observed with GlyCl silencing (*Figure 1—figure supplement 2C,E*). Unlike the GlyCl model, we did not see retraction bulbs in the EC of Kir2.1 mice, but did observe evidence of axon damage in the DG at later time points. Iba1 immunostaining peaked between 13 and 16 dpi in both EC and DG of Kir-expressing animals, consistent with neuronal damage in these areas (*Figure 1—figure supplement 2D,F*). Control animals received tTA-dependent YFP virus without Kir and showed no loss of fluorescence nor any change in Iba1 beyond the needle track in either the DG or EC over 4 weeks post injection (data not shown).

## Neural populations neighboring EC2 are resistant to silencing

Examining the effects of GlyCl activation on other brain regions would allow us to test whether silencing is generally toxic or whether distinctive properties of EC2 neurons confer vulnerability to silencing. The neuropsin-tTA driver line used to express GlyCl in EC2 is also active in connected regions of the presubiculum and parasubiculum, which allowed us to test whether electrical silencing elicited cell death outside of EC2 (*Yetman et al., 2015*; *Figure 2A and B*). We first confirmed that GlyCl expression was sufficient to suppress neuronal firing in pre- and parasubiculum using whole-cell recordings in acute brain slices. Action potential firing was significantly reduced in YFP+ neurons in both regions upon bath application of 100 nM IVM to the same level as EC2 at this dose (*Figure 2C and D*, *Figure 2—figure supplement 1*). We counted YFP+ cells in each brain area before and after systemic IVM treatment at 5 mg/kg (1× IVM). As before, a single dose of IVM significantly reduced the number of labeled EC2 cells in GlyCl+ mice (*Figure 2E*). In contrast, IVM caused no loss of pre- or parasubiculum neurons at this dose, suggesting that these neighboring regions may not share EC2's vulnerability to activity disruption (*Figure 2E*). We next tested whether cell loss in either of these regions would be unmasked at a higher dose of IVM. We posited that higher doses of IVM allow for increased GlyCl activation and more effective electrical silencing (*Lerchner et al., 2007*; *Lynagh and Lynch, 2010*). We treated an independent set of mice with a single injection of 10 mg/kg IVM (2× IVM). This dose caused significantly more cell loss in the EC2 than 1× IVM (70% at 2× vs. 50% at 1×, *Figure 2E*). We also found that parasubiculum neurons that had been resistant to 1× IVM now showed mild cell loss at 2× IVM (*Figure 2E*). Nevertheless, the proportion of neurons lost in parasubiculum at 2× IVM (30%) was considerably smaller than in EC2 at 1× IVM (50%). Remarkably, neuron counts in presubiculum were not altered at either dose. The simplest explanation for these differences in vulnerability would be differences in GlyCl expression level between the three areas. We therefore measured GlyCl levels in individual cells from each region using immunofluorescence to detect transgenic human glycine receptor (*Figure 2F–H*). We found that the presubiculum expressed relatively low levels of GlyCl receptor, and therefore we could not exclude the possibility that its resilience was a result of low transgene expression (*Figure 2I*). In contrast, GlyCl expression was practically identical in EC2 and parasubiculum, despite their distinct patterns of cell loss (*Figure 2I*). We were

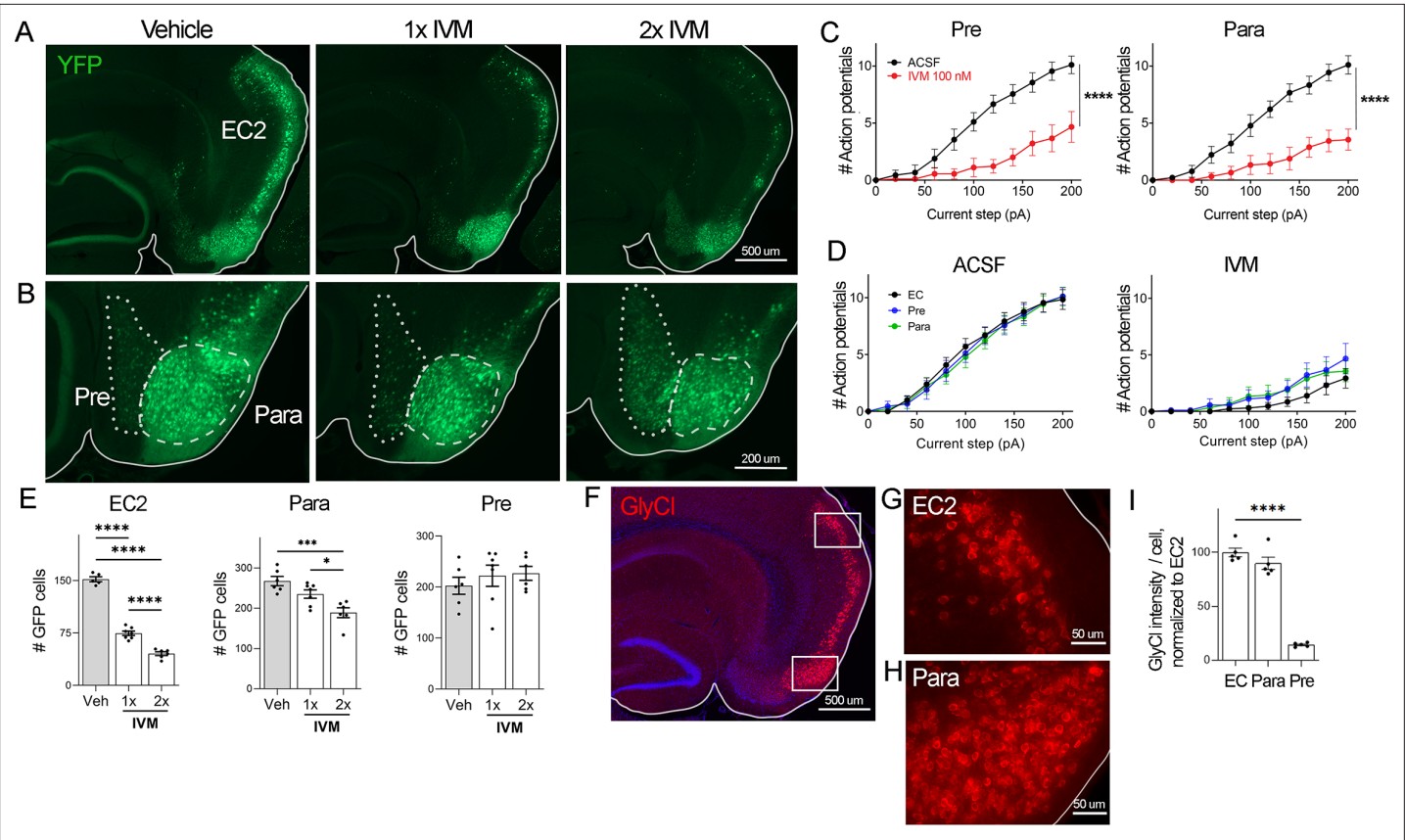

**Figure 2.** EC2 is more vulnerable to electrical inactivation than neighboring regions. (**A, B**) The Nop-tTA line drives expression of transgenic GlyCl in neural populations beyond the entorhinal cortex (EC), including the pre- and parasubiculum. Bigenic Nop-GlyCl mice were injected with vehicle or ivermectin (IVM) at 5 mg/kg (1×) or 10 mg/kg (2×) to assess the relationship between dose and neuronal loss in these brain regions. (**C**) Whole-cell recordings of pre- and parasubiculum show that action potentials initiation in both cell types is effectively suppressed upon 100 nM IVM exposure, similar to EC2. Data expressed as mean ± SEM, two-way repeated-measures ANOVA (p-value for interaction of treatment × current). Pre n = 9 cells from nine slices (four mice); para n = 9 cells from nine slices (four mice). (**D**) Graphs showing action potential number as a function of current step in EC, parasubiculum, and presubiculum were overlaid to show that neurons in all three brain regions were similarly silenced at 100 nM IVM (mean ± SEM, two-way repeated-measures ANOVA with Sidak's post-test). The data shown here for comparison of the three regions also appears separately in *Figure 1A*, (**C**), and *Figure 2—figure supplement 1*. (**E**) Quantification of YFP+ cells revealed that only EC2 neurons are lost at a dose of 1× IVM, and this population is severely affected by 2× IVM. Parasubiculum neurons only become significantly vulnerable to silencing at 2× IVM. Presubiculum neurons showed no cell loss at either dose. (**F**) Immunostaining for the transgenic human glycine receptor revealed that EC2 and parasubiculum neurons express high levels of GlyCl, while expression in presubiculum was considerably lower. (**G, H**) High-magnification images of GlyCl immunostaining from (**E**) show that fluorescence intensity was similar in EC2 and parasubiculum. (**I**) Quantification of per-cell fluorescence intensity confirms that GlyCl expression is nearly identical between EC2 and parasubiculum despite differences in cell loss upon IVM silencing. Graphs depict average number of YFP+ cells (**E**) or average per-cell intensity of GlyCl+ neurons (**I**) for each animal ± SEM, one-way ANOVA with Tukey's post-test. Cell counts were done 14 dpi. Veh n = 5, 1× IVM n = 7, 2× IVM n = 6. *p<0.05, ***p<0.001, ****p<0.0001. Associated with *Figure 2—figure supplement 1*.

The online version of this article includes the following source data and figure supplement(s) for figure 2:

**Source data 1.** Numerical data for *Figure 2*.

**Figure supplement 1.** Brain regions outside entorhinal cortex (EC) express GlyCl in the Nop-GlyCl model and are electrically suppressed by ivermectin (IVM).

**Figure supplement 1—source data 1.** Numerical data for *Figure 2—figure supplement 1*.

surprised to discover this decoupling of silencing, GlyCl expression, and cell loss. While activity in all three brain regions was efficiently silenced by IVM, EC2 and parasubiculum expressed higher levels of GlyCl protein, and EC2 alone saw substantial cell loss. These findings suggest that EC2 neurons are especially sensitive to activity perturbation, and considerably more vulnerable to electrical inactivation than cells in neighboring pre- and parasubiculum.

## Reelin-positive stellate cells are the primary EC2 population lost upon silencing

Neuronal silencing only killed a fraction of the GlyCl-expressing EC2 neurons, raising the possibility of differential vulnerability within this population. EC2 contains multiple neuronal subtypes distinguished by their molecular markers, firing properties, morphology, and connectivity. Axons from stellate cells in the medial EC and fan cells in the lateral EC connect with DG through the perforant pathway, and we predicted this population would be preferentially lost in our Nop-Gly mice given the disappearance of axon terminals from the DG following EC2 silencing. The three main cell types within EC2 can be distinguished by expression of specific protein markers: excitatory stellate and fan cells are characterized by reelin expression, excitatory pyramidal neurons by Wfs1, and inhibitory neurons by GABAergic enzymes (*Kitamura et al., 2014*; *Kitamura et al., 2015*). We found that the Nop-tTA driver line is active in both reelin-positive stellate cells and Wfs1-positive pyramidal neurons, but completely absent from inhibitory neurons (*Figure 3A* and data not shown). Consistent with past work, we found YFP labeled roughly half of the reelin- and Wfs1-expressing EC2 neurons (*Yasuda and Mayford, 2006*). We harvested Nop-GlyCl mice 7 days after a single dose of vehicle or IVM (5 mg/kg) and co-immunostained for YFP, reelin, and Wfs1. Of the two cell types, only reelin-expressing cells decreased after IVM, while the number of Wfs1-positive cells did not change significantly (*Figure 3B and C*). This data suggests that reelin + stellate cells are preferentially vulnerable to cell death after silencing. This conclusion coincides with the fact that the parasubiculum, comprised predominantly of Wfs1+ cells, was resistant to neuronal silencing (*Figures 2E and 3D*), and with recent work suggesting that expression of Wfs1 is protective in the EC during AD (*Chen et al., 2022*).

## EC2 neurons are also vulnerable to synaptic silencing

Neuronal silencing with GlyCl or Kir2.1 affects both action potential initiation and synaptic transmission. We next wanted to tease apart whether one or both effects were required for cell loss. To do this, we used virally expressed tetanus toxin (TeTX) to suppress neurotransmitter release from EC2 neurons in adult Nop-tTA mice without affecting electrical activity. Co-expressed YFP was used to visualize the silenced neurons and YFP alone was injected as a control (*Figure 4A*). Field recording in acute slices from virally injected mice confirmed that TeTX expression significantly reduced electrical transmission between EC2 and DG and that the extent of suppression was consistent with the percentage of neurons expressing the virus (*Figure 4B*). Expression of TeTX-YFP initially labeled both EC2 neuronal cell bodies and their axon terminals in the DG, similar to the pattern in YFP-injected control mice. Viral expression peaked at 7 dpi and shortly afterward YFP intensity began to fade in the DG of TeTX mice (*Figure 4C*). Loss of perforant path labeling progressed over the next 2 weeks until nearly undetectable. Animals injected with YFP alone showed no change in labeling over the same time period. Loss of axonal labeling in TeTX mice was accompanied by the emergence of fractin staining in the EC that was specific to mice expressing TeTX (*Figure 4D*). Fractin-positive cells first appeared at 10 dpi, became more prominent at 13 dpi, and largely abated by 19 dpi. As with GlyCl-based silencing, we observed reactive microglia in the DG and EC as an early marker of impending cell loss (*Figure 4E*) and found sporadic cells labeled with active caspase 3 (*Figure 4F*). This cascade of events leading from impaired neurotransmission to cell death was comparable across the various silencing methods we tested, suggesting that EC2 neurons are broadly susceptible to inactivity, and that inhibiting synaptic release alone is sufficient to cause EC2 cell loss.

## TeTX and GlyCl elicit distinct cell death processes

In all cases of EC2 silencing, we observed the loss of DG innervation over time. Closer inspection of the GlyCl and TeTX-expressing mice revealed a more nuanced story. Although electrical suppression by GlyCl caused gradual loss of DG fluorescence, synaptic inhibition by TeTX caused pronounced axonal disintegration within the DG (*Figure 5*). GlyCl silencing was characterized by the appearance of retraction bulbs in deep layers of EC, suggesting neuronal loss followed intact axonal withdraw. No retraction bulbs were visible in TeTX-treated animals; instead, the axonal processes became fragmented with small bright puncta that were reminiscent of Wallerian degeneration. This qualitative distinction between TeTX and GlyCl systems warranted further investigation of possible mechanistic differences caused by silencing at the cell body vs. the synapse.

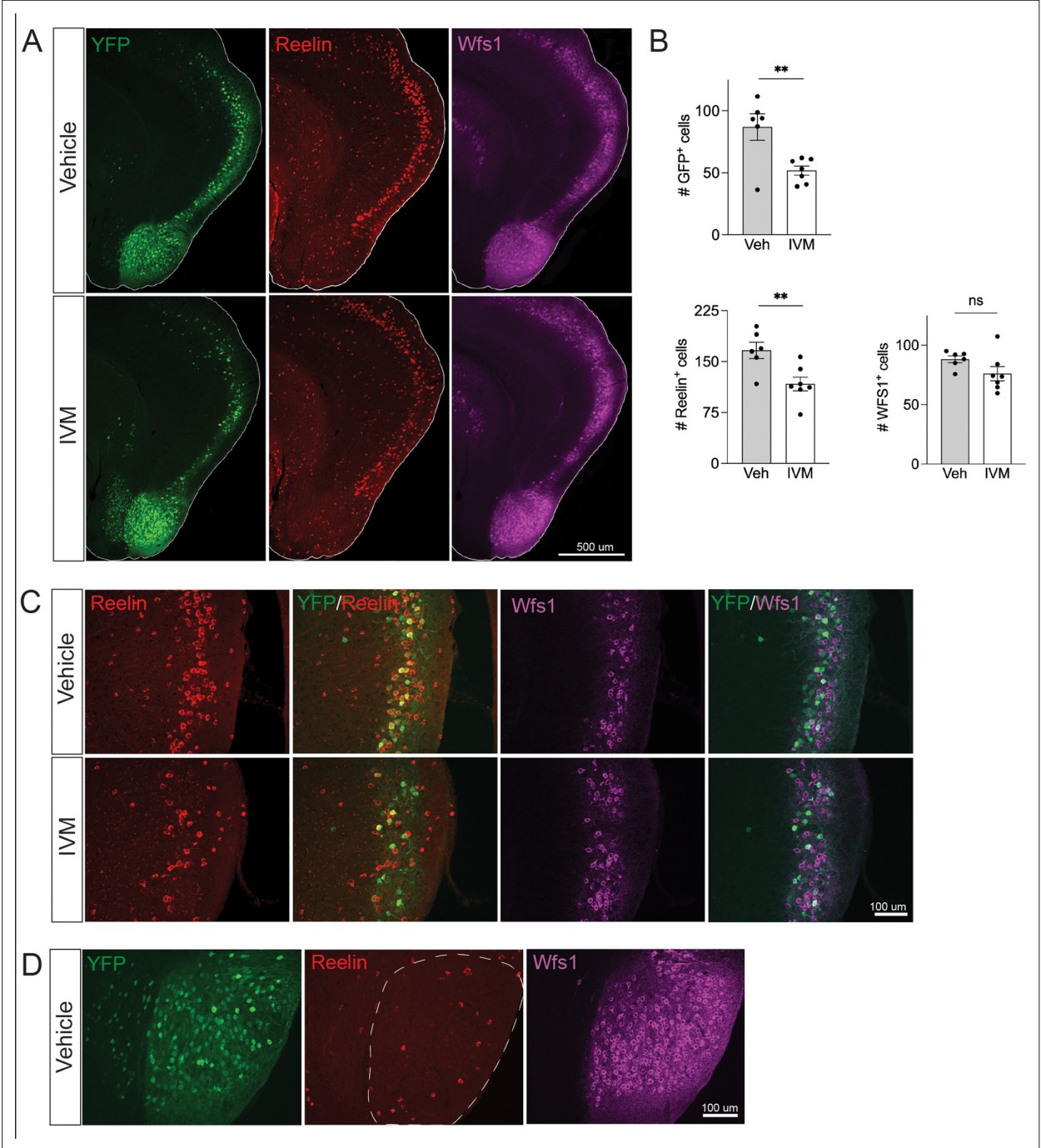

**Figure 3.** Electrical silencing preferentially kills reelin-positive stellate cells in EC2. (**A**) Co-immunostaining of Nop-GlyCl mice for YFP, reelin, and Wfs1 revealed that the transgene was expressed in both EC2 reelin+ stellate cells and Wfs1+ pyramidal neurons. Animals were harvested 7 days after a single injection of vehicle or ivermectin (IVM) (5 mg/kg). (**B**) EC2 YFP+ cells were diminished by ~40% upon silencing, reelin cells were reduced by ~30%, and Wfs1 cells were preserved, suggesting preferential vulnerability of stellate neurons upon electrical silencing. Data expressed as mean ± SEM. Veh n = 6, IVM n = 7. Student's *t*-test (two-sided). **p<0.01. (**C**) High-magnification images of reelin, Wfs1, and YFP immunostaining in EC2 show a loss of reelin+ and YFP+ cells but no difference in Wfs1+ neurons after electrical silencing. (**D**) High-magnification images of the parasubiculum show that cells in this region are predominantly Wfs1+, consistent with their resistance to activity perturbation.

The online version of this article includes the following source data for figure 3:

**Source data 1.** Numerical data for *Figure 3*.

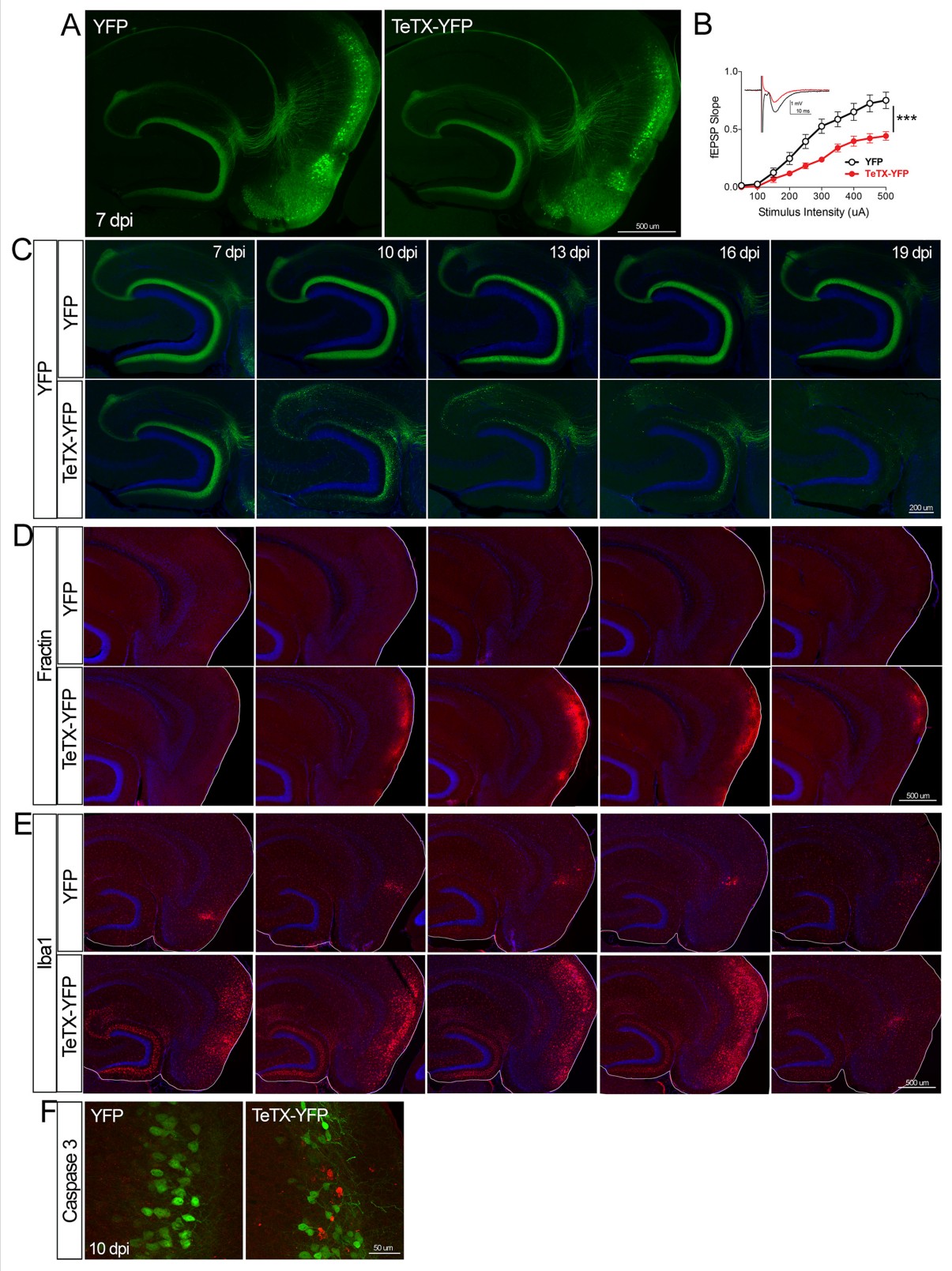

**Figure 4.** Entorhinal cell death is also elicited by synaptic silencing with tetanus toxin (TeTX). (**A**) AAV was used to express either YFP or YFP plus TeTX in entorhinal neurons. By 7 dpi, axons terminating in the dentate gyrus (DG) can be clearly seen in both constructs by native fluorescence. (**B**) Virally delivered TeTX effectively diminishes synaptic transmission into DG. Data expressed as mean ± SEM, two-way repeated-measures ANOVA (p-value for interaction of condition × stimulus intensity, ***p<0.001). n = 15 slices from three mice (YFP), 17 slices from three mice (TeTX), harvested 7–9 dpi. Note

*Figure 4 continued on next page*

*Figure 4 continued*

that YFP data shown here is identical to *Figure 1—figure supplement 2B* as the same control group was used for both experiments. (**C**) TeTX causes disintegration of labeled perforant path axons. No change in dentate innervation is seen in animals injected with YFP control virus. DAPI counterstain indicates the dentate granule cell layer. (**D**) Fractin immunostaining appears in EC as axons disintegrate in DG. (**E**) Microglial activation detected with Iba1 appears rapidly in both DG and EC by 7 dpi in mice injected with AAV-TeTX and persists in EC for more than a week before resolving. Microglial activation is seen only at the injection site in mice injected with AAV-YFP. (**F**) Active caspase 3 immunostaining (red) is found among YFP+ cells in the EC (green) only in animals co-expressing TeTX. 4 dpi n = 2 YFP, 2 TeTX; 7 dpi n = 4 YFP, 4 TeTX; 10 dpi n = 3 YFP, 4 TeTX; 13 dpi n = 5 YFP, 4 TeTX; 16 dpi n = 6 YFP, 4 TeTX; 19 dpi n = 3 YFP, 5 TeTX; 22 dpi n = 2 YFP. Unnecessary cerebellar tissue, where present, was masked in Photoshop to match fields of view across panels.

The online version of this article includes the following source data for figure 4:

**Source data 1.** Numerical data for *Figure 4*.

One important feature of the models described so far is that they allowed us to manipulate a targeted subset of EC2 neurons. The Nop-tTA driver line used to express GlyCl, Kir2.1, and TeTX is active in approximately 40% of EC2 neurons (*Yasuda and Mayford, 2006*). The remaining EC2 neurons were unaffected by our genetic and viral manipulations. As a result of this incomplete penetrance, our experiments created a situation whereby some neurons continued to fire normally while adjacent cells were inactivated. During postnatal development, competition between active and inactive neurons is used to refine the perforant path (*Yasuda et al., 2011*). TeTX silencing of the same neuronal subset caused axonal loss in the developing brain that was qualitatively similar to what we saw in the adult. Yasuda et al. further showed that axonal loss could be prevented by eliminating the competition between active and inactive cells with tetrodotoxin (TTX). This led us to wonder whether competition might account for neuronal loss upon silencing in the adult EC.

To test this hypothesis, we infused TTX or saline just dorsal to the right EC starting 3 days after Nop-tTA mice were injected with TRE-TeTX-YFP virus. TTX infusion was continued until harvest 7 days

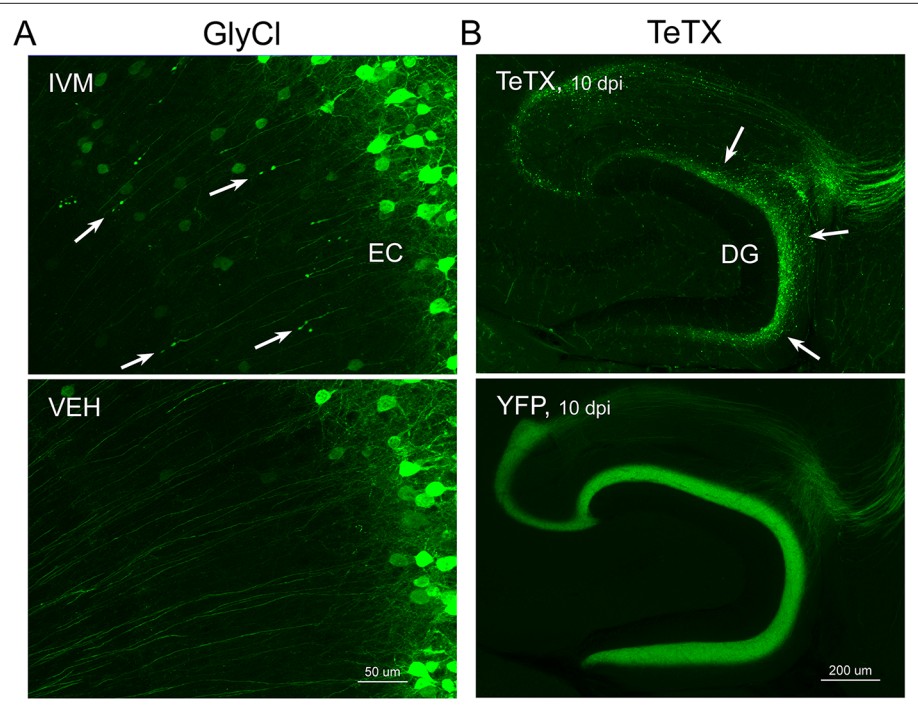

**Figure 5.** Different patterns of axonal damage attend neuronal silencing by GlyCl and tetanus toxin (TeTX). (**A**) Retraction bulbs were visible in deep entorhinal cortex (EC) layers of Nop-GlyCl bigenic animals 4 days after injection of ivermectin (IVM) (upper panel), while axons in vehicle-treated mice remain intact (lower panel). (**B**) In contrast to the discrete retraction bulbs seen with GlyCl, TeTX silencing caused axonal fragmentation throughout the dentate gyrus (DG) molecular layer (arrows, upper panel). YFP labeling in TeTX mice became punctate and innervation deteriorated within 10 days after viral injection, while the axons of EC neurons labeled with YFP alone remained robust and bright (lower panel).

later (10 dpi after viral injection). By injecting slightly dorsal, TTX reached the EC and DG but these areas were spared from damage. The spread of TTX and efficacy of EC silencing were verified by the loss of c-fos labeling following chemically induced seizures (*Figure 6—figure supplement 1*). Although we would have liked to test extended time points in the degenerative process initiated by TeTX, TTX treatment beyond 7 days caused locomotor impairment as it diffused beyond the target site. Consistent with our initial experiments, control mice injected with TeTX virus but infused with saline showed pronounced axonal disintegration visible as bright YFP puncta in the DG at 10 dpi (*Figure 6A*). In contrast, axonal labeling remained robust and intact in mice infused with TTX. Preservation of axonal innervation was limited to the infused hemisphere; degeneration on the contralateral side was identical to that seen in animals infused with saline (*Figure 6B*). These differences were reflected in both the width of YFP labeling and the intensity of YFP fluorescence measured at the crest of the DG (*Figure 6C and D*). Consistent with the axon preservation, the extent of microglial activation normally observed in the DG of TeTX mice was also abated by TTX infusion (*Figure 6B and E*). In conclusion, TTX infusion preserved axonal innervation and maintained microglial homeostasis in mice expressing TeTX, suggesting that degeneration due to synaptic silencing may be governed by ongoing competition between active and inactive contacts.

We next tested whether the same competitive mechanism was responsible for cell death after GlyCl-based silencing. We began TTX infusion 12 hr before injecting IVM to ensure TTX had reached the EC at the time of GlyCl activation. TTX infusion was continued for another 2.5 dpi after IVM injection to suppress neuronal firing throughout the period when GlyCl might remain active (*Zhao et al., 2016*). Animals were euthanized 7 days after IVM to examine whether widespread inactivity caused by TTX would prevent cell death in the subset of EC2 neurons that were silenced with GlyCl. Contrary to our results with TeTX, infusion of TTX had no effect on neuronal survival in the GlyCl model. We found no difference in the area of YFP fluorescence within EC2 between mice infused with TTX and saline (*Figure 6—figure supplement 2A*). Both IVM-treated groups lost ~66–75% of the YFP+ area compared to vehicle-injected mice, similar in magnitude to EC2 cell counts in IVM-treated GlyCl mice without cannulation (see *Figure 1B*). TTX infusion also had no effect on fractin expression or microglial activation evoked by IVM (*Figure 6—figure supplement 2B and C*). Importantly, TTX infusion caused no cell loss in control mice injected with vehicle instead of IVM, indicating that TTX infusion by itself does not cause cell death. Given this negative outcome, we tested two modifications of the infusion procedure to rule out the possibility of incorrect TTX targeting or insufficient TTX duration. Neither 3 days of TTX infusion targeting the DG nor 7 days of infusion targeting dorsal EC changed the outcome (data not shown). These findings indicate that cell death in the GlyCl model cannot be explained by competition between the silenced cells and their active neighbors, and that separate mechanisms govern neuronal death following electrical versus synaptic silencing.

## Discussion

Our efforts to model entorhinal dysfunction due to AD led us to three novel discoveries about this circuit. First, we demonstrate that mature EC2 neurons require ongoing activity for survival long beyond the postnatal critical period. This activity dependence distinguishes EC from neighboring pre- and parasubiculum, which are largely resilient to activity changes. Second, we demonstrate that electrical versus synaptic silencing can elicit distinct forms of axonal damage prior to cell death. Finally, we show that axonal disintegration due to synaptic silencing is governed by competition between active and inactive neurons, suggesting that this circuit may maintain an activity-dependent critical period well into adulthood.

### Activity-dependent neuronal survival persists in the adult cortex

We show that both electrical silencing due to chloride or potassium flux and synaptic silencing due to arrest of neurotransmitter release are each sufficient to cause axonal damage of EC2 neurons followed by cell body demise. Such wholesale changes in activity-dependent circuit patterning are traditionally thought to be limited to a brief window during postnatal development (*Murase, 2014*; *Wong and Marín, 2019*). We suspect that our discovery arises at least in part from three fortuitous aspects of our experimental system: (1) the prolonged suppression provided by the manipulations we chose compared with the shorter duration tools used in earlier work, (2) the decision to look at time points

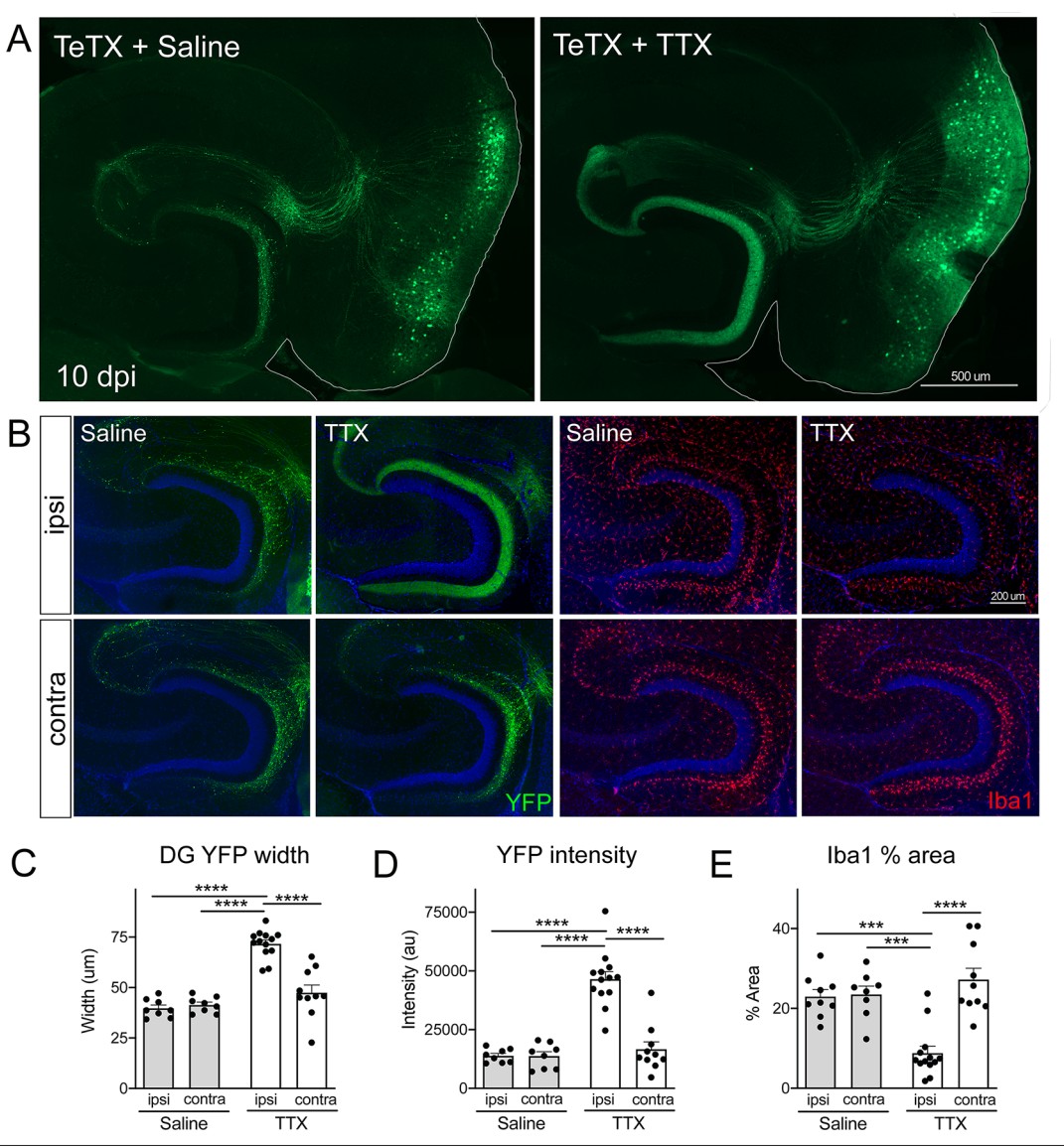

**Figure 6.** Activity-dependent competition governs entorhinal cortex (EC) axonal disintegration due to tetanus toxin (TeTX) silencing. Mice expressing TeTX + YFP in EC neurons were treated by local infusion of tetrodotoxin (TTX) or saline. Unilateral infusion targeting dorsal EC began 3 days after viral injection and continued until mice were harvested 7 days later (10 dpi). (**A**) Representative images showing the spread of TeTX transduction in the EC as visualized by native YFP fluorescence. Preservation of dentate gyrus (DG) innervation can be seen in the TTX-infused animal (right), in contrast to the punctate 'beads on a string' labeling of remaining perforant path axons in the saline-infused condition (left). (**B**) Left panels: images show YFP labeling of TeTX-expressing axons (green) in the DG both ipsilateral and contralateral to the infusion cannula. DG innervation is only preserved ipsilateral to TTX infusion. Right panels: images show Iba1 immunostaining in the DG, where TeTX-evoked microglial activation is prevented in the TTX-infused hemisphere. (**C, D**) The width (**C**) and intensity (**D**) of YFP labeling at the crest of the DG confirms axonal sparing ipsilateral to TTX infusion. (**E**) Analysis of % area occupied by Iba1 in the DG confirms that microglial activation was prevented in the TTX-infused hemisphere. Graphs depict average values for each animal ± SEM, one-way ANOVA with Tukey's post-test (missed injection target for some hemispheres precluded two-way ANOVA). n = 8–9 (saline infusion), 10–13 (TTX infusion). *p<0.05, ***p<0.001, ****p<0.0001. Associated with *Figure 6—figure supplements 1 and 2*.

The online version of this article includes the following source data and figure supplement(s) for figure 6:

**Source data 1.** Numerical data for *Figure 6*.

*Figure 6 continued on next page*

*Figure 6 continued*

**Figure supplement 1.** Broad silencing of neuronal activity in the entorhinal cortex (EC) following tetrodotoxin (TTX) infusion.

**Figure supplement 2.** Global suppression of entorhinal cortex (EC) activity does not prevent cell death by GlyCl activation.

**Figure supplement 2—source data 1.** Numerical data for *Figure 6—figure supplement 2*.

well after silencing had subsided, and (3) our focus on the EC-DG connection where circuit architecture, a layer-specific driver line, and a co-expressed cytosolic YFP label allowed clear visualization of axonal degeneration and cell loss.

Numerous studies have used chemical, chemogenetic, or optogenetic tools to silence entorhinal input to the DG without observing signs of axonal damage or cell death (*Rueckemann et al., 2016*; *Miao et al., 2015*; *Ormond and McNaughton, 2015*; *Rowland et al., 2018*; *Robinson et al., 2017*; *Kanter et al., 2017*; *van Wijngaarden et al., 2020*; *Heys et al., 2020*; *Qin et al., 2018*; *Rodriguez et al., 2020*). Nearly all of these studies tested the electrophysiological consequences of entorhinal dysfunction on spatial coding and spatial navigation behaviors. Changes in place cell properties were detected immediately after entorhinal inactivation and animals were generally harvested upon completion of the task. We delayed harvesting the initial Nop-GlyCl mice until many days after IVM injection to map the pharmacodynamics of extended silencing with the GlyCl system. Further, optogenetic or DREADD chemogenetic tools display temporal precision on the scale of milliseconds to minutes, but silencing with GlyCl and TeTX is slow and sustained. The half-life for IVM in the brain is over 13 hr and behavioral impairments persist for several days after IVM administration (*Zhao et al., 2016*; *Lerchner et al., 2007* and data not shown). The effect of TeTX is even more protracted, in theory silencing transmission as long as expression continues. While neither the GlyCl or TeTX would be ideal for experiments requiring restoration of physiology or behavior after silencing, both were well suited to our initial purpose of modeling circuit impairments due to neurodegenerative disease.

A fundamental question raised by our findings is the threshold beyond which cell death pathways are engaged by prolonged or complete silencing. We identified one previous study that tested prolonged EC inactivation using hM4Di DREADD with chronic CNO infusion over 6 weeks of study (*Rodriguez et al., 2020*). This paper is most similar to our experiments in its sustained manipulation of EC activity, but focused on downstream tau and amyloid pathology in the hippocampus rather than cell survival or axonal morphology in the EC. Immunostaining suggested that many DREADD-expressing neurons survived 6 weeks of CNO exposure. The divergent outcomes between chronic partial suppression with DREADDs and shorter but complete silencing with GlyCl or TeTX supports the possibility of a threshold dividing survival and death.

## Preferential resilience of Wfs1 cells within EC2

Our transgenic driver line fortuitously allowed us to silence a portion of both major excitatory cell types within EC2. This arrangement provided a direct test of vulnerability between the reelin+ stellate cells that project to DG and the Wfs1+ pyramidal neurons that project to CA1 and contralateral EC. Of the two populations, we found that pyramidal neurons were relatively resilient to activity perturbation, while stellate cells were preferentially vulnerable. This outcome is consistent with recent work showing that Wfs1 expression protects EC2 neurons from tauopathy and degeneration (*Chen et al., 2022*). Tau tangles appear in dense patches within human EC2 (*Braak and Braak, 1995*; *Van Hoesen et al., 1986*; *Van Hoesen et al., 1991*), where, in contrast to the mouse EC, islands are composed of reelin+ stellate cells (*Witter et al., 2017*). Both Wfs1+ and reelin+ cells are lost from EC2 during AD; however, the loss of reelin appears to be much more profound while more than half of Wfs1+ cells survive late into the disease (*Chen et al., 2022*; *Chin et al., 2007*; *Delpech et al., 2021*). Synaptic markers are also lost early in AD from the terminal field of EC2 neurons in DG, lending further support to the preferential vulnerability of hippocampal-projecting stellate and fan cells (*Hamos et al., 1989*; *Scheff et al., 1996*; *Scheff et al., 2006*). The factors contributing to differential vulnerability between stellate and pyramidal neurons are only just being elucidated, but the distinct patterns of cell loss we found within EC2 suggest that molecular signatures may be just one piece of this puzzle. Our data suggest that connectivity and function within the hippocampal-EC circuit may be equally important in governing differential neuronal vulnerability.

## Different modes of silencing elicited distinct patterns of axonal damage

We unexpectedly discovered that the same population of hippocampal-projecting EC2 neurons could adopt different patterns of axonal degeneration prior to cell death. Axons silenced by GlyCl displayed evidence of simple axonal retraction beginning within 1–2 days of silencing onset. These axons remained intact, although displaced from the DG, with bulb-like structures at their tips. This pattern was strikingly different from the overt axonal disintegration observed after TeTX silencing. Atrophy in TeTX-expressing axons was reminiscent of Wallerian degeneration, which is characterized by axon blebbing and fragmentation. Both patterns of axon elimination are found in the developing CNS, although rarely in the same cell population (*Neukomm and Freeman, 2014*; *Riccomagno and Kolodkin, 2015*). The developing thalamocortical projection stands out in displaying both retraction and disintegration, with the choice seeming to depend on the length of axon segment being remodeled (*Neukomm and Freeman, 2014*; *Portera-Cailliau et al., 2005*). Axon length did not differ in our model as the same projections underwent degeneration under each form of silencing. Instead, our models differed by the method used to prevent neurotransmission. GlyCl suppresses electrical activity by membrane hyperpolarization (*Zhao et al., 2016*; *Lerchner et al., 2007*) or reduced membrane conductance (*Weir et al., 2017*), either of which would dampen evoked neurotransmitter release onto postsynaptic DG targets while having no effect on spontaneous release. In contrast, TeTX has no direct effect on electrical activity, but can potently inhibit both evoked and spontaneous neurotransmitter release by preventing vesicle fusion at the membrane (*Schiavo et al., 1992*; *Humeau et al., 2000*). We hope that future studies will identify the distinct signaling pathways activated by these different means of neuronal silencing.

## Neuronal competition regulates axon degeneration due to TeTX but not cell loss due to GlyCl

Competition determining which cells live or die has been well-studied during the development of sensory circuits (*Blakemore, 1991*; *Yu et al., 2004*; *Zhao and Reed, 2001*). The same competitive pressure was recently discovered during axonal refinement of the developing perforant pathway (*Yasuda et al., 2011*). Yasuda et al. found that infusing TTX into the EC during development prevented the loss of axons silenced by TeTX. This elegant experiment suggested that competition between active and inactive axons drove developmental refinement of the perforant path. We used a similar approach to determine whether competition might also drive axonal degeneration and cell death in the adult EC. The divergent outcome of this manipulation in TeTX and GlyCl models surprised us, but supported the idea that synaptic silencing and electrical suppression evoke distinct mechanisms of cell death in the adult EC. Local infusion of TTX preserved axonal integrity in neurons expressing TeTX but had no effect on cell loss in neurons silenced by GlyCl. Electrical silencing by GlyCl appears to kill EC2 neurons by a noncompetitive, possibly cell-intrinsic mechanism, but axonal disintegration due to synaptic silencing by TeTX appears to be influenced by activity in neighboring neurons. This competitive mechanism – and the experiments used to demonstrate it – is consistent with the process used during developmental refinement of the perforant path; however, we cannot rule out the alternative explanation that TTX may act directly on the TeTX-expressing cells rather than through their neighbors. Our experiments also stop short of demonstrating that axonal preservation in TeTX-expressing mice is sufficient to protect the cell body due to the limited duration of TTX infusion. Nevertheless, this discovery raises the possibility that the critical window for EC circuit refinement – potentially including wholesale neuron elimination – remains open long into adulthood. This discovery also raises the question of whether increasing synaptic activity might be neuroprotective for EC neurons. Several groups have tested EC activation in the adult brain using chemo- or optogenetic tools; however, most evoked only short bursts of activity that may not be sufficient to induce a survival imbalance (*Zhang et al., 2013*; *Zhang et al., 2014*; *Kanter et al., 2017*; *Leung et al., 2018*).

## Limitations of our experimental models

We also must address limitations of our model systems that raise important caveats to several of our conclusions. First and foremost, our studies focused almost exclusively on EC2. We do not know whether our chemogenetic tools would evoke axonal damage and cell death in other areas of the brain. We sought to address this concern by measuring cell loss in neural populations outside EC that were also silenced in the Nop-GlyCl model. We found no evidence of cell loss in pre- or parasubiculum

at IVM doses sufficient to kill ~50% of silenced EC2 neurons. Even at higher doses of IVM, these areas remained relatively preserved compared with EC2. This data suggests that EC2 is particularly vulnerable to activity changes, but our comparison did not look beyond these three areas.

We also appreciate that the inter-animal variability in viral transduction prevented us from performing meaningful cell counts to assess cell loss after silencing with Kir or TeTX. Our conclusions instead relied on loss of DG innervation and the appearance of damage markers such as Iba1 and fractin. In addition, stereotaxic viral injections likely reached a smaller fraction of EC neurons than transgenic expression of GlyCl, which would alter the proportion of active vs. silenced cells in each system. This could influence the level of competition within each system, which could alter the degenerative phenotype we observed. Nevertheless, TeTX expression always produced axon disintegration rather than retraction, regardless of transduction efficiency. We recognize that the proportion of silenced cells might also have affected the potential for rescue by broad TTX silencing. Similarly, axonal disintegration in TeTX mice was consistently rescued by TTX, independent of the number of cells transduced. Because of this, we conclude that the differences in degenerative phenotypes between GlyCl and TeTX are most likely intrinsic to the silencing system, but appreciate the need for additional studies using matched viral or transgenic expression systems.

We also note that other teams have used TeTX silencing in the mossy fiber and Schaffer collateral pathways for up to 6 months with no evidence of lasting axonal damage (*Lopez et al., 2012*; *Nakashiba et al., 2012*; *Nakashiba et al., 2008*). Of course, our experiments placed TeTX in EC2 neurons, while the Tonegawa studies expressed TeTX in DG and CA3 neurons. We do not know whether DG and CA3 are more resilient than EC2; however, our studies of pre- and parasubiculum suggest that EC2 may be more vulnerable to activity disruption than other nearby parts of the hippocampal-entorhinal circuit.

## Vulnerability of the EC-hippocampal circuit may derive from its function

Unlike the primary sensory cortices that precisely map their input during a defined critical period, the EC remains highly plastic throughout life and is constantly modified during memory formation. While both EC and neighboring parasubiculum support spatial awareness and navigation, EC integrates that information to support long-term memory, whereas parasubiculum primarily updates the internal spatial model without contributing to spatial memory (*Tang et al., 2016*). This need to remain continuously plastic may render the EC more vulnerable than nearby regions to insults that cause an activity imbalance between neurons, extending and offering a specific target for the idea that homeostatic collapse may be a driving force in neurodegeneration (*Styr and Slutsky, 2018*; *Frere and Slutsky, 2018*). Consistent with this idea, EC cells that degenerate early in AD are characterized by expression of axonal remodeling and excitability genes, suggestive of heightened plasticity (*Leng et al., 2021*; *Roussarie et al., 2020*). Here, we experimentally impaired transmission between the EC and hippocampus, but several naturally occurring insults may have similar consequences. For example, pathological tau accumulation in EC neurons might impair the initiation or propagation of action potentials in affected cells while neighboring neurons remain normally active. EC2 is particularly prone to tangle pathology, where these lesions mark the earliest stages of AD (*Hyman et al., 1984*; *Van Hoesen et al., 1986*; *Van Hoesen et al., 1991*; *Braak and Braak, 1991*). Alternatively, the formation of amyloid plaques in the DG could physically disrupt the communication between pre- and postsynaptic cells. Plaques are prominent in the molecular layer of the dentate, precisely where the presynaptic terminals of the perforant path connect with postsynaptic dendrites of granule cells (*Crain and Burger, 1988*; *Geddes et al., 1986*; *Hyman et al., 1986*; *Braak and Braak, 1991*). Importantly, in the early stages of pathology formation, these communication deficits would be localized to individual neurons with tangles or their axons terminating near plaques. The altered microenvironment of these affected neurons could establish a competitive situation in which those neurons would be disadvantaged relative to their neighbors, perhaps leading to their demise. This theory offers a new model by which the pathologies of AD may underlie the early vulnerability of EC neurons. Moreover, these studies raise the possibility that lifelong plasticity needed for episodic memory may be a disadvantage in disease.

## Methods

### Mice

Animals were housed in accordance with the NIH Guide for the Care and Use of Laboratory Animals and were 3–6 months of age at the time of treatment. Littermates were randomly distributed between experimental groups. Mice of both sexes were used for all experiments. Group sizes were based on prior studies; power analysis was not done. Biological replicate values (individual animals) are indicated in the figure legends; technical replicate values (number of sections analyzed per animal for each stain) are indicated 'Imaging and image quantitation'.

Nop-tTA Line S (also known as tTA-EC Prss19 [*Yasuda and Mayford, 2006*], MMRRC strain #31779) and TetO-GFP-nls-LacZ (also known as nls-lac-CMK; *Mayford et al., 1996*) were gifts from Mark Mayford, Scripps Research Institute. The lines were maintained on a C57BL/6J background.

TRE-GlyCl-YFP Line 9531 (*Zhao et al., 2016*; Jax strain #29301) was described previously and was maintained on an FVB/NJ background for more than 10 generations.

Nop-GlyCl and their single transgenic Nop-tTA siblings for these experiments were derived from the intercross of male Nop-tTA × female TRE-GlyCl breeders (F1 FVB;B6 background).

Nop-tTA mice used for viral injections were derived from the intercross of male Nop-tTA x female FVB/NJ wild-type breeders (Jax #1800) or from the single transgenic offspring of male Nop-tTA x female TRE-GlyCl breeders (both resulting in an F1 FVB;B6 background).

### Viral constructs

Three tTA-dependent pAAV expression vectors were created for study, one encoding YFP alone, one encoding tetanus toxin light chain (TeTX) with YFP, one encoding Kir with YFP. pAAV-TRE3G-T2a-YFP contained the TRE3G promoter from pTRE3G (#631168, Clontech, Mountain View, CA), followed in turn by a modified multiple cloning site (BamHI-MfeI-AgeI-NruI-SpeI), the *Thosea asigna* virus 2A sequence, the enhanced yellow fluorescent protein (YFP) from pEYFP-N1 (#6006-1, Clontech), the woodchuck hepatitis virus posttranscriptional regulatory element (WPRE, from pAAV2-GluClα-WPRE; *Lerchner et al., 2007*), and a bovine growth hormone (BGH) polyadenylation signal. The entire insert was flanked by inverted terminal repeats (ITR) from AAV2.

Tetanus toxin light chain (TeTx$_{LC}$) was amplified by PCR from pGEMTEZ-TeTx$_{LC}$ (Addgene #32640; *Yu et al., 2004*) using forward primer: GCGCGCAATTGGCCACCATGCCGATCACCATCAACAACTT and reverse primer: GCGCGACTAGTAGCGGTACGGTTGTACAGGTT. The resulting PCR product was digested with MfeI and SpeI and ligated into the MfeI and SpeI sites of pAAV-TRE3G-T2a-YFP to build pAAV-TRE3G-YFP-T2a-TeTx$_{LC}$.

Kir was amplified by PCR from pCAG-Myc-Kir2.1$^{E224G/Y242F}$-2A-EGFP (similar to pCAG-Kir2.1-T2A-tdTomato, Addgene #60598, kind gift of Mingshan Xue, *Xue et al., 2014*) using forward primer: GCGCGGGATCCGCCACCATGGAGCAGAAGCT and reverse primer: GCGCGACTAGTTCCACTGCCTATCTCCGATTC. The resulting PCR product was digested with BamHI and SpeI and ligated into the BamHI and SpeI sites of pAAV-TRE3G-T2a-YFP to build pAAV-TRE3G-YFP-T2a-Kir$^{E224G/Y242F}$.

Viral plasmids created here will be made available upon request to researchers at nonprofit institutions under MTA from the Baylor College of Medicine.

### Viral packaging

Packaging and purification of viral constructs in AAV8 was performed by the Gene Vector Core at Baylor College of Medicine, as described previously (*Huichalaf et al., 2019*; *Park et al., 2021*).

### IVM administration

GlyCl-expressing bigenic or trigenic mice and controls were administered with either vehicle solution (60:40 propylene glycol:0.9% saline) or 5 mg/kg IVM (1% Ivomec, Merial, Lyon, France, or 1% Noromectin, Norbrook, Newry, Northern Ireland) diluted to 2 mg/ml in vehicle via intraperitoneal injection. Mice were sacrificed for histological assessment at time points ranging from 6 hr to 30 days after treatment, as indicated. For 2× IVM treatment, animals received 10 mg/kg IVM at 4 mg/ml.

## Stereotaxic viral delivery

Adult Nop-tTA mice on an F1 FVBB6 background underwent surgery between 3 and 5 months of age. Mice were fixed in a stereotaxic frame, administered buprenorphine, ketoprofen, and local lidocaine/bupivacaine for analgesia, and anesthetized with isoflurane. Bilateral craniotomies were made using a 0.45 mm drill bit. Craniotomies targeting the EC were positioned at ML ± 3.60 mm, while AP coordinates were adjusted for each animal using the bregma-lambda distance (right hemisphere: AP = 0.808 × BL distance + 1.636 mm; left hemisphere: AP = 0.829 × BL distance + 1.46 mm). For EC expression, AAV8-TRE3G-YFP, AAV8-TRE3G-TeTX$_{LC}$-YFP, or AAV8-TRE3G-Kir2.1-YFP was injected bilaterally using a 31-gauge stainless steel needle (#22031-01, Hamilton Co., Reno, NV) attached to a microsyringe injector (#UMC4, World Precision Instruments, Sarasota, FL). The needle was slowly lowered to DV –3.2 mm and remained in place for 10 min after the injection. Each hemisphere was injected with 300 nl virus ($6.9 \times 10^8$ genome copies total for YFP and TeTX, empirically equivalent to $5.7 \times 10^8$ gc for Kir2.1) at a rate of 50 nl/min.

## Optimizing TTX dosage, targeting, and spread

TTX (Abcam, Cambridge, MA, ab120054) or 0.9% saline containing 0.04% Trypan blue for visual confirmation of catheter flow was preloaded into a micro-osmotic pump (#1003D, Alzet, Cupertino, CA) and primed by immersion in sterile 0.9% saline overnight at 37°C. Mice were anesthetized with isoflurane, and an indwelling 30-gauge cannula (#0008851, Alzet) cemented to the skull (Kerr, Orange, CA, 34417) at AP –4.5, ML +3.0, and DV –2.5 mm that targeted immediately above the right EC or at AP –3.1, ML +3.0, and DV –2.65 to target the dorsal DG. Immediately following cannula placement, the primed micro-osmotic pump was positioned subcutaneously through a mid-scapular incision and the catheter routed subcutaneously to connect with the cannula. The pump flow rate of 1 µl/hr was chosen to deliver 24 µl/day TTX. To measure the spread and efficacy of TTX, mice were injected with the convulsant drug pentylenetetrazol (PTZ, 25 mg/kg) 90 min prior to harvest and killed at 6–48 hr after implantation to examine the spread of dye and the extent of c-fos labeling induced by PTZ. TTX fully suppressed neuronal activity caused by PTZ at 12 hr after the start of infusion, and this timing was used for IVM injection during competition experiments.

## TTX administration

TTX (2.3 uM) or 0.9% saline containing 0.04% Trypan blue was prepared and infused as described above to target right dorsal EC. Where indicated, mice received a single i.p. injection of IVM (5 mg/kg) or vehicle given 12 hr after completion of cannulation surgery. For Nop-GlyCl mice, the osmotic pump was removed 3 days after implantation and animals were harvested 7 days after IVM or vehicle injection. For TeTX mice, the osmotic pump was implanted 3 days after viral injection and replaced 4 and 7 days later to continue TTX administration until animals were harvested 10 days after viral injection.

## Tissue harvest and sectioning

Mice were killed by pentobarbital overdose and transcardially perfused with phosphate-buffered saline (PBS) followed by 4% paraformaldehyde (PFA) in PBS. The brain was removed and postfixed by immersion in PBS containing 4% PFA at 4°C overnight, and then cryoprotected by immersion in 30% sucrose in PBS at 4°C until equilibrated. Tissue was sectioned in the horizontal plane at 35 µm thickness using a freezing-sliding microtome. Sections were stored in cryoprotectant media at –20°C until use.

## Immunolabeling and histology

### Fractin and YFP, fluorescence

For labeling of fractin with or without co-detection of YFP, free-floating sections were rinsed in TBS and then endogenous peroxide was quenched by incubating sections 30 min at room temperature (RT) with 0.9% H$_2$O$_2$ in TBS with 0.3% Triton X-100 and 0.05% Tween 20. Sections were washed in TBS, blocked with 5% normal goat serum in TBS with 0.3% Triton X-100 and 0.05% Tween 20 for 1 hr at RT, and then incubated in primary antibody at 4°C for 72 hr (1:1000 chicken anti-GFP, Abcam, ab13970; 1:5000 rabbit anti-fractin, Phosphosolutions, 592-FRAC). Sections were washed several times in TBS, followed by incubation overnight at 4°C in secondary antibody diluted in block (1:500 goat anti-chicken Alexa 488, Life Technologies, A11039; 1:5000 biotinylated goat anti-rabbit, Vectastain Elite

ABC HRP kit, PK-6101, Vector Laboratories, Burlingame, CA). Sections were washed several times in TBS with 0.3% Triton X-100 and 0.05% Tween 20, incubated for 30 min at RT with Vectastain components A and B diluted one drop each into 30 mL in TBS, and then washed overnight in TBS. Sections were incubated for 10 min in Cy3 Plus Amplification Reagent diluted 1:300 in 1× Plus Amplification Diluent (TSA Plus Cyanine 3 System, PerkinElmer, Waltham, MA, NEL744001KT), washed several times in TBS with 0.3% Triton X-100 and 0.05% Tween 20, incubated 10 min in 0.2 ug/ml DAPI diluted in TBS, and then washed again in TBS.

### Cleaved caspase 3 and YFP, fluorescence

Sections were rinsed with TBS and blocked with TBS containing 0.1% Triton X-100 (TBST) plus 5% normal goat serum for 1 hr at RT before 3-day incubation in primary antibody diluted in block (1:1000, chicken anti-GFP, Abcam, ab13970; 1:200 rabbit anti-cleaved caspase 3, EMD Millipore, AB3623). Sections were washed several times in TBS, followed by overnight incubation at 4°C in secondary antibody diluted 1:500 in block (donkey anti-rabbit Alexa 568, Life Technologies, A10042). Sections were again washed in TBS. Tissues were then incubated overnight at 4°C in tertiary antibody diluted in block (goat anti-chicken Alexa 488, Life Technologies, A11039, and goat anti-donkey Alexa 568, Life Technologies [discontinued]), and washed a final time in TBS.

### Iba1 and YFP, fluorescence

For fluorescence labeling of Iba1 with or without co-detection of YFP, sections were rinsed with TBS and then blocked with TBST plus 5% normal goat serum for 1 hr at RT before overnight incubation in primary antibody diluted 1:1000 in block (rabbit anti-Iba1, Wako Chemicals USA, Richmond, VA, 019-19741; 1:1000 chicken anti-GFP, Abcam). Sections were washed several times in TBS, followed by 2 hr incubation at RT in secondary antibody diluted 1:500 in block (goat anti-rabbit Alexa 568, Life Technologies; 1:500 goat anti-chicken Alexa 488, Life Technologies). Sections were again washed in TBS, incubated 10 min in 0.2 ug/ml DAPI diluted in TBS before being mounted and coverslipped.

### GlyCl and YFP, fluorescence

Sections were rinsed with TBS and then blocked with TBST plus 5% normal goat serum for 1 hr at RT before overnight incubation in primary antibody diluted 1:1000 in block (rabbit anti-hu Glycine Receptor alpha 1, Novus Biologicals, Centennial, CO, NB300-113; 1:1000 chicken anti-GFP, Abcam). Sections were washed several times in TBS, followed by a 2 hr incubation at RT in secondary antibody diluted 1:500 in block (goat anti-rabbit Alexa 568, Life Technologies A11011; 1:500 goat anti-chicken Alexa 488, Life Technologies A11039). Sections were incubated 10 min in 0.2 ug/ml DAPI diluted in TBS before being washed in TBS, mounted and coverslipped.

### YFP, reelin, and Wfs1, fluorescence

For this experiment only, Nop-Gly mice were fed rodent chow containing 200mg/kg doxycycline starting at postnatal day 14 (Bio-Serv, S3888). Animals were removed from doxycycline chow at six weeks of age and expressed GlyCl for 6-10 weeks before IVM or vehicle treatment. A 1-in-6 series of sections was washed in TBS and blocked in TBS containing 0.1% Triton-X plus 5% normal donkey serum for 1 hr at RT. Sections were then incubated overnight at 4°C in primary antibody diluted in block (1:1000 chicken anti-GFP, Abcam, ab13970; 1:400 goat anti-reelin, Invitrogen, PA5-47537; 1:1000 rabbit anti-Wfs1, ProteinTech, 26995-1-AP). Sections were washed with TBS followed by 2 hr at RT in secondary antibody diluted 1:500 in block (donkey anti-chicken Alexa 488, Jackson ImmunoResearch Laboratories, 703-545-155; donkey anti-rabbit Alexa 647, Life Technologies, A-32795; donkey anti-goat Alexa 568, Life Technologies, A-11057). 0.2 ug/ml DAPI was added during the last 10 min of secondary antibody incubation before being washed in TBS, mounted and coverslipped.

### c-fos, fluorescence

Sections were washed in TBS and blocked in TBS containing 0.1% Triton-X plus 5% normal goat serum for 1 hr at RT. Sections were then incubated overnight at 4°C in primary antibody diluted in block (1:500 rabbit anti-cFos, Millipore, ABE457). Sections were washed with TBS followed by 2 hr at RT in

secondary antibody diluted 1:500 in block (goat anti-rabbit 568, Life Technologies, A11011). Sections were washed in TBS, mounted and coverslipped.

All sections were mounted onto Superfrost Plus slides (Fisher Scientific, Pittsburgh, PA) and coverslipped with ProLong Diamond anti-fade mounting media (P36970, Invitrogen/Life Technologies Corp, Eugene, OR).

### Iba1, colorimetric

Sections were rinsed with TBS before endogenous peroxide was quenched by incubating sections for 30 min at RT in TBS with 0.9% hydrogen peroxide and 0.01% Triton-X-100. Sections were washed in TBS and then blocked with TBS containing 0.1% Triton X-100 (TBST) plus 5% normal goat serum for 1 hr at RT before overnight incubation in primary antibody diluted 1:1000 in block (rabbit anti-Iba1, Wako Chemicals USA). Sections were washed several times in TBS, followed by overnight incubation at 4°C in biotinylated secondary antibody diluted 1:500 in block (biotinylated goat anti-rabbit, Vectastain Elite ABC HRP kit, PK-6101, Vector Laboratories). Sections were washed in TBS and then incubated for 1 hr at RT in Vectastain kit components A and B, each diluted one drop each into 10 ml in TBS. Sections were again washed and then developed for 1 hr in DAB solution (Sigmafast D4418 tablets dissolved in 50 ml water). Sections were washed in TBS, mounted on Superfrost Plus slides, and allowed to dry overnight. Slides were processed through alcohol series (70, 95, and 100%) to xylene and then coverslipped with Permount (Fisher Scientific).

### β-Galactosidase stain

Bigenic Nop-tTA/tetO-LacZ and trigenic Nop-tTA/tetO-LacZ/TRE-GlyCl tissues were harvested and sectioned as above, except that brains were postfixed by PFA immersion for 2 hr prior to immersion in 30% sucrose in PBS. A 1-in-6 series of sections was washed in TBS, mounted onto Superfrost Plus slides, and allowed to dry for 24–72 hr. Slides were rehydrated in solution A (2 mM $MgCl_2$ in PBS) for 35 min, incubated in prewarmed solution B (2 mM $MgCl_2$, 0.2% NP40, 0.1% sodium deoxycholate in PBS) for 1 hr, and then incubated in prewarmed solution C overnight (solution B with 10 mM ferro/ferri cyanide, 0.6 mg/ml X-gal). Slides were rinsed in stop solution (0.1% Triton-X 100, 1 mM EDTA in HEPES buffered saline, pH 7), postfixed overnight with 4% PFA, washed in PBS, and then air dried for 1 hr. Slides were dehydrated through alcohol series (70, 95, and 100%) to xylene, followed by rehydration to water. Slides were counterstained in nuclear fast red, dehydrated again through the alcohol series to xylene, and finally coverslipped with Permount.

## Imaging and image quantification

Figure images have been taken at exposure times that optimize the display histogram and then adjusted in Photoshop to visually match the background levels across animals. Unnecessary cerebellar tissue where present was masked in Photoshop to match field of view across panels. All images used for quantitation within each experiment were imaged at the same exposure and adjusted identically for analysis.

No data points were excluded from analyses; however, animals in which the viral injection was not correctly targeted were excluded from analysis based on spatial distribution of fluorescent protein that served as a viral reporter. In some cases (*Figure 6*), one hemisphere was correctly targeted while the other was not. In these cases, only the correctly targeted hemisphere was included in analysis for that animal. The experimenter was blinded to treatment for cell counts in *Figure 3*. Blinding was not done for other analyses because in many cases, the phenotype of the tissue revealed the treatment of the animal.

### Fractin, YFP+ cells, and YFP+ retraction bulbs

A 1-in-6 series of horizontal sections through the entire dorsoventral extent of the brain were co-immunostained for YFP and fractin as described above. Tiled images were acquired using a Zeiss Axio Scan.Z1 at ×10 magnification (Carl Zeiss AG, Oberkochen, Germany). Exposure time and lamp intensity were constant for all sections. Sections used for analysis were plane-matched across animals. Regions of interest (ROIs) were identified from anatomical landmarks, and objects within each ROI were manually counted using the multipoint tool and ROI manager in ImageJ/Fiji Mac v2.0.0. For Nop-GlyCl animals without TTX treatment (*Figure 1*), YFP+ and fractin+ cells were counted in EC

layer 2 from four horizontal sections spaced at ~300 μm intervals spanning –2.68 to –3.60 mm from bregma (*Franklin and Paxinos, 2008*). For Nop-GlyCl and Nop-tTA + TeTX animals with TTX treatment (*Figure 6*, *Figure 6—figure supplement 2*) 6–8 sections spanning –2.68 to –3.60 mm from bregma were used for analysis and all fractin+ cells were manually counted using both hemispheres for analysis. Retraction bulbs were counted in the deep layers of EC (below layer 2) in four horizontal sections across the same interval (*Figure 1*). Graph data expressed as average values per section for each animal ± SEM.

### β-Galactosidase

A 1-in-6 series of horizontal sections through the dorsoventral extent of the brain was stained for lacZ as described above. Tiled images were acquired using a Zeiss Axio Scan.Z1. ROIs were identified using anatomical landmarks and blue-labeled nuclei within the ROI were manually counted using the multipoint tool and ROI manager in ImageJ/Fiji Mac v2.0.0. For the EC, labeled nuclei were counted bilaterally in all sections spanning the entire EC2 (6–8 sections per mouse). Graph data expressed as average % area for each animal ± SEM.

### Iba1 and YFP % area

A 1-in-6 series horizontal sections stained for Iba1 or GFP was scanned using a Zeiss Axioscan Z.1 and imported into Image J2 ([Fiji Is Just] ImageJ 2.0.0-rc-68/1.52g; Java 1.8.0_172 [64 bit]). From these, 3–4 plane-matched sections spanning the medial EC were chosen for analysis. Each image was converted to 8-bit and three ROIs were manually outlined: superficial EC covering layers 2–3, molecular layer of the DG including both blades, and the medial zone of stratum radiatum in CA1. For each stain, a threshold was first determined using the Yen algorithm on the CA1 ROI as a control region to identify the background intensity for that section. The background value from CA1 was applied as the threshold for signal the EC or DG ROI and the % area above threshold was calculated.

### GlyCl intensity

To measure GlyCl staining intensity as shown in *Figure 2*, ROIs were drawn around each of EC2, pre, and parasubiculum in Fiji. After converting to 16-bit and processing the image using the 'subtract background' function, a threshold was applied within each of the region-specific ROIs using the Otsu algorithm. A selection was created based off the binary image to create a mask of the GlyCl signal in each region. The GlyCl mask area was then applied to the original 16-bit image and integrated density was measured for each ROI to return the total intensity across the entire region. To normalize the integrated density per cell, original images were again converted to 16-bit, background subtracted, then a Gaussian blur filter was applied to generalize the shape of the cells. For analyzing presubiculum neurons, no blur was applied. The Otsu threshold was then applied using the ECII, pre- and parasubiculum ROIs drawn earlier. The watershed function was applied to separate joined cells. To count cells, particles were analyzed within each ROI. Integrated density was then divided by the total number of particles per region and divided by the EC2 measurement to generate an EC2 normalized intensity per cell. Data shown is the average intensity per cell from four plane-matched sections per animal.

### YFP intensity, DG width

For the TeTX + TTX experiment shown in *Figure 6*, the width of YFP labeling was measured across the crest of the DG using the measure feature in Fiji. YFP immunofluorescence intensity was measured using the integrated density function of Fiji from a 50 μm × 50 μm region within the crest of the DG (IntDen = area × mean gray value). Regional values were measured from each hemisphere individually in three plane-matched sections per animal and averaged to calculate YFP intensity. For the Kir experiment shown in *Figure 1—figure supplement 2*, YFP immunolabeling within the DG was manually outlined and measured using the mean gray value of Fiji (MGV = sum of pixel values/number of pixels). This measure accounts for differences in the size of selected ROIs. Regional values were measured from two plane-matched sections per animal and averaged to calculate YFP intensity.

## YFP+, reelin+, and WFS1+ cells

A 1-in-6 series of horizontal sections through the dorsoventral extent of the brain were co-immunostained for YFP, reelin, and Wfs1 as described above. Tiled images were acquired using a Zeiss Axio Scan.Z1 at ×20 magnification (Carl Zeiss AG). Exposure time and lamp intensity were consistent across all sections. Sections used for analysis were plane-matched across animals. An ROI was drawn around EC2 and added to the ROI manager in Fiji. Images were converted to 8-bit before initializing the cell counter to manually count YFP+, reelin+, or WFS1+ cells within each ROI using the multipoint tool in Fiji. Cells were counted from three horizontal sections spaced at ~300 µm intervals spanning –2.52 to –2.84 mm from bregma (*Franklin and Paxinos, 2008*). Graph data expressed as average cell number per section for each animal ± SEM.

## Brain slice preparation and electrophysiology

Brain slices from Nop-GlyCl or virally injected Nop-tTA mice (YFP, TeTX-YFP, or Kir-YFP) were prepared as described previously with minor modifications (*Li et al., 2015*; *Li et al., 2017*; *Zhao et al., 2016*). Mice were deeply anesthetized with isoflurane and the brains were rapidly removed and placed in N-methyl-D-glucamine (NMDG)-based 'cutting buffer' containing (in mM) 80 NMDG, 2.5 KCl, 0.5 $CaCl_2$, 10 $MgSO_4$, 1.2 $NaH_2PO_4$, 30 $NaHCO_3$, 25 D-dextrose, 5 sodium ascorbate, 2 thiourea, 3 sodium pyruvate, pH 7.4 adjusted with HCl and 300–310 mOsm (*Ting et al., 2014*). Coronal or horizontal slices containing EC, presubiculum, or parasubiculum (300–320 µm) were cut in 95% $O_2$- and 5% $CO_2$-oxygenated cutting buffer. Slices were incubated at 35°C in a submerged chamber containing artificial cerebrospinal fluid (ACSF) equilibrated with 95% $O_2$ + 5% $CO_2$-oxygenated buffer for at least 30 min and maintained at RT afterward until transfer to a recording chamber. The ACSF contained (in mM) 126 NaCl, 2.5 KCl, 2.4 $CaCl_2$, 1.2 $MgCl_2$, 1.2 $NaH_2PO_4$, 21.4 $NaHCO_3$, 11.1 D-dextrose, pH 7.4, and the osmolarity was adjusted to 300–310 mOsm.

Whole-cell patch-clamp recordings were made from visually identified YFP-labeled or unlabeled EC layer 2, presubiculum, or parasubiculum neurons with infrared DIC and fluorescence optics (Axio-Examiner, Zeiss). Recording electrodes were pulled from borosilicate glass (1.5 mm × 0.86 mm diameter) on a P-1000 horizontal micropipette puller (Sutter Instruments) to a resistance of 4–5 MΩ when filled with the intracellular solution. Current-clamp experiments used an intracellular pipette solution containing (in mM) 120 K gluconate, 20 KCl, 0.1 EGTA, 4 $MgCl_2$, 5 NaCl, 10 HEPES, 4 MgATP, 3 NaGTP, 10 phosphocreatine, pH 7.2, 290 mOsm. Data were low-pass filtered at 5 kHz and sampled at 10 kHz using Axon MultiClamp 700 B amplifier and Digidata 1440A data acquisition system under control of pClamp 10.7 software (Molecular Devices). Recordings measured the number of action potentials generated in response to 250 ms stepped current injections (–60 pA to 200 pA at 20 pA increments). Throughout the experiment, access resistance (Ra) was 10–20 MΩ and cells were discarded if Ra drifted above 20%. Series resistance and cell capacitance were compensated. Current–response relationships were constructed by counting the number of action potentials generated at each current step in the presence or absence of 100 nM IVM. IVM was allowed to perfuse for 10–15 min before post-IVM recordings were made. All recordings were made in ACSF at an ambient temperature of 32–34°C. One neuron was recorded per slice and 4–5 slices were recorded per mouse. Results compared pre- and post-IVM for each neuron.

In a separate set of experiments, extracellular field potentials were recorded using electrodes filled with ACSF described above. Excitatory postsynaptic potentials (fEPSPs) were recorded in the DG medial molecular layer following perforant path stimulation through a twisted nichrome wire electrode. Stimulus–response curves were calculated by plotting initial slope of the fEPSPs against stimulus intensity (50–500 uA in 25 uA steps). All recordings were made at ambient temperature of 32–34°C.

The experimenter was blinded to treatment/genotype for all electrophysiology experiments.

## Statistical analysis

Data comparisons were done using two-tailed unpaired Student's *t*-test for two group comparisons, or one-, two-, or three-way ANOVA for three or more group comparisons, followed by Dunnett's (one-way for comparison to control), Tukey's (one-way for comparison across all groups), or Sidak's (two-way repeated-measures for comparison to control and three-way for comparison across all groups)

post-test. Statistical analyses and graphing of data were done using Prism 6.0, 8.4, or 9.3 (GraphPad, La Jolla, CA). All graphs display group means ± SEM.

## Acknowledgements

We thank Jennifer Saldana, M Danish Uddin, Shaina Gong, and Rebecca Corrigan for assistance with animal care and genotyping; Mingshan Xue and Daoyun Ji for valuable experimental advice; and Mark Mayford and Karsten Baumgartel for sharing the Nop-tTA and nls-lac-CMK mouse lines. AAV preparation was done with the help of Kazu Oka and the BCM Gene Vector Core. Confocal imaging was done with the help of Jason Kirk at the BCM Optical Imaging and Vital Microscopy Core. This work was funded by NIH RF1 AG058188, RF1 AG058188-01S1, RF1 AG054160, and R01 NS092615 to JLJ, R25 GM056929 and HHMI Gilliam Fellowship GT13620 (support for GAP), F31 AG067676 to CAW, Alzheimer's Association fellowship AARF-17-533487 and BrightFocus Foundation fellowship A2015016F to SDG. Slide scanning was done with the help of Cecilia Ljungberg at the BCM RNA In Situ Hybridization Core, supported by NIH shared instrumentation grant 1S10OD016167 and NIH IDDRC grant U54HD083092 from the Eunice Kennedy Shriver National Institute of Child Health & Human Development.

## Additional information

### Funding

| Funder | Grant reference number | Author |
|---|---|---|
| National Institute on Aging | RF1 AG058188 | Joanna L Jankowsky |
| National Institute on Aging | RF1 AG058188-01S1 | Joanna L Jankowsky |
| National Institute on Aging | R01 NS092615 | Joanna L Jankowsky |
| Howard Hughes Medical Institute | GT13620 | Joanna L Jankowsky |
| National Institute on Aging | F31 AG067676 | Caleb A Wood |
| Alzheimer's Association | AARF-17-533487 | Stacy D Grunke |
| BrightFocus Foundation | A2015016F | Stacy D Grunke |

The funders had no role in study design, data collection and interpretation, or the decision to submit the work for publication.

### Author contributions

Rong Zhao, Melissa Comstock, Formal analysis, Investigation, Visualization, Methodology; Stacy D Grunke, Caleb A Wood, Conceptualization, Formal analysis, Funding acquisition, Investigation, Visualization, Methodology, Writing – review and editing; Gabriella A Perez, Formal analysis, Funding acquisition, Investigation, Visualization, Methodology; Ming-Hua Li, Formal analysis, Investigation, Visualization; Anand K Singh, Formal analysis, Investigation; Kyung-Won Park, Resources, Investigation; Joanna L Jankowsky, Conceptualization, Resources, Supervision, Funding acquisition, Visualization, Writing – original draft, Project administration, Writing – review and editing

### Author ORCIDs

Caleb A Wood (iD) http://orcid.org/0000-0002-3320-0485
Joanna L Jankowsky (iD) http://orcid.org/0000-0002-5593-2310

### Ethics

Animals were handled and housed in accordance with recommendations in the NIH Guide for Care and Use of Laboratory Animals. All animal procedures were reviewed and approved by the BCM IACUC under protocol AN-4975.

### Decision letter and Author response

Decision letter https://doi.org/10.7554/eLife.83813.sa1

Author response https://doi.org/10.7554/eLife.83813.sa2

## Additional files

### Supplementary files
• MDAR checklist

### Data availability
All data generated or analysed during this study are included in the manuscript and supporting files.

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
