## [Editor Report]

This is a fundamental study that demonstrates that ongoing neuronal activity plays a key role in the vulnerability of specific neuronal cell types in layer 2 of the entorhinal cortex that communicates with the hippocampus. The authors provide compelling evidence that chronic silencing of excitatory neurons in the entorhinal cortex leads to their degeneration. Reelin-positive stellate neurons were the most vulnerable to silencing. The authors propose that developmental mechanisms associated with activity-dependent circuit refinement could be aberrantly reactivated in the context of Alzheimer's disease.

---

## [Decision Letter]

**Decision letter after peer review:**

Thank you for submitting your article "Activity disruption causes degeneration of entorhinal neurons in a mouse model of Alzheimer's circuit dysfunction" for consideration by *eLife*. Your article has been reviewed by 2 peer reviewers, and the evaluation has been overseen by a Reviewing Editor and John Huguenard as the Senior Editor. The following individual involved in the review of your submission has agreed to reveal their identity: Dori Derdikman (Reviewer #2).

Essential revisions:

The reviewers and editors found your results to be fundamental and solid. Please address the suggestions of reviewer #1 in the revised version of the manuscript.

*Reviewer #1 (Recommendations for the authors):*

– No obligation but I thought that the authors could cite somewhere one or the two previews from Inna Slutsky in Neuron and Nature Neuro in 2018 since, in my opinion, these papers are some of the most crystal-clear models of how synaptic homeostasis dysregulation and Alzheimer's disease are interconnected.

– Would it be possible to obtain evidence for the effective silencing of mEC2 neurons at the moment when retraction bulbs form in the GlyCl model? I am just wondering if the high expression of Hcn1 in mEC2 neurons might have unexpected effects when IVM starts to wash off. If neurons start to experience rebound firing at that time, we would just want to make sure that the retraction bulbs are not appearing at that time because of that, but that they started when the neurons were still very hyperpolarized. – I was wondering why the electrophysiology data post-Kir2.1 overexpression was field recordings after perforant path activation as opposed to say whole-cell recordings of mEC2 neurons. Could the effect seen in supplementary Figure 2 B be due to the onset of axon loss?

– I might have missed it in the main text, but I am wondering if the observation about the absence of axon retraction bulbs in the legend of Suppl. 2 should not be in the main text. I finished my first reading of the paper with the impression that only the Tetanus toxin experiment was not showing the same retraction bulbs.

– It would be great to show the c-fos stainings post TTX infusion and seizure induction. Was c-fos staining eliminated in the DG or in ECII or both? In the LECII?

– When thinking of the differences between the TeTx and GlyCl, I was wondering if you could maybe give a sense of the proportion of mECII cells that were targeted by the stereotaxic injection? Is it the entirety of mECII that is transduced? Do you see any difference in the density/thickness of the MML between the GlyCl YFP and the TeTx-YFP before and after degeneration and at different dorso-ventral levels? I do not think that this is essential data, but since you are making the comparison between inhibition of degeneration by TTX in both paradigms, these might represent at least a discussion point that could explain some discrepancies. i.e. more or less competition in one case vs. the other?

– In the part "Preferential resilience of Wfs1 cells within EC2" in the discussion, I am not sure I understand "but the changeable pattern of cell loss we found between neighboring regions and within EC2 suggests that molecular signatures must be just one part of a complex balance": do you imply that these neighboring cells are pretty close in terms of molecular signatures, and so that molecular signatures are not the only determinant of vulnerability? Pyramidal and stellate cells of mECII do have a number of molecular differences that could account for their differential vulnerability including Wfs1 expression as you mention. But maybe I misunderstood the sentence.

– In the last section of the discussion, the authors say "For example, formation of neurofibrillary tangles in EC neurons might impair…". I am wondering if the authors should not replace NFT with a broader "tau pathology" or "pathological tau accumulation", considering the fact that there starts to be considerable evidence that neurofibrillary tangles are not necessarily the toxic tau species dysregulating neuronal excitability.

*Reviewer #2 (Recommendations for the authors):*

I have read the paper and have no further requests. I think it is good to go.

---

## [Author Response]

Reviewer #1 (Recommendations for the authors):– No obligation but I thought that the authors could cite somewhere one or the two previews from Inna Slutsky in Neuron and Nature Neuro in 2018 since, in my opinion, these papers are some of the most crystal-clear models of how synaptic homeostasis dysregulation and Alzheimer's disease are interconnected.

We thank the reviewer for reminding us about Dr. Slutsky's elegant work and thoughtful summaries on how circuit dysfunction can promote disease pathogenesis. We now include these citations in both the Introduction and Discussion sections of the manuscript:

Introduction, second paragraph:

“Increased neuronal activity promotes further amyloid release, creating a vicious cycle (Cirrito et al. 2005; Styr and Slutsky 2018; Frere and Slutsky 2018).”

Discussion, final paragraph:

“This need to remain continuously plastic may render the EC more vulnerable than nearby regions to insults that cause an activity imbalance between neurons, extending and offering a specific target for the idea that homeostatic collapse may be a driving force in neurodegeneration (Styr and Slutsky 2018; Frere and Slutsky 2018).”

– Would it be possible to obtain evidence for the effective silencing of mEC2 neurons at the moment when retraction bulbs form in the GlyCl model? I am just wondering if the high expression of Hcn1 in mEC2 neurons might have unexpected effects when IVM starts to wash off. If neurons start to experience rebound firing at that time, we would just want to make sure that the retraction bulbs are not appearing at that time because of that, but that they started when the neurons were still very hyperpolarized.

The reviewer raises an excellent question about whether rebound firing affects the formation of retraction bulbs. Our data suggests that retraction bulbs form within the first 24 hours after IVM injection, which is within the window that IVM remains in the brain (the half-life for IVM in brain = 13.3 hr, see Zhao et al., Cell Rep. 2016, PMID 27373150, Figure 3B). Although unpublished, we have preliminary behavioral data suggesting that the suppressive effect of a single IVM injection endures for at least 2 days, well after retraction bulbs first appear. We appreciate the apparent paradox this data presents in demonstrating that an entorhinal-dependent cognitive function could recover just as neuronal silencing evokes cell death in the entorhinal circuit. We believe this paradox is resolved by the difference between the relatively larger number of cells that are silenced by IVM due to the wider limbic expression of the tTA driver line (including EC plus presubiculum, parasubiculum, and retrosplenial cortex) and the considerably smaller number of EC-specific neurons that are lost in the aftermath.

**Author response image 1. sa2fig1:** Acquisition of conditioned fear takes several days to recover from entorhinal silencing. Contextual fear conditioning was performed in 3 month old animals trained at varying times following a single injection of IVM. Recall for the trained chamber was measured 24 hours after conditioning as % time immobile during a 5 min test session. IVM treatment had no effect at any time on control animals lacking GlyCl expression (white bars), while associative memory was significantly impaired in animals with entorhinal GlyCl trained 3 hours or 2 days after IVM injection, and finally recovered to control levels by 3 days after IVM dosing.

We have modified the first section of Results entitled "*Unexpected loss of EC2 neurons following chloride-based neuronal silencing*" with the following addition:

“Retraction bulbs began while IVM was still active in the brain, suggesting their emergence was the result of neuronal silencing and not due to rebound activity (R. Zhao et al. 2016)”

– I was wondering why the electrophysiology data post-Kir2.1 overexpression was field recordings after perforant path activation as opposed to say whole-cell recordings of mEC2 neurons. Could the effect seen in supplementary Figure 2 B be due to the onset of axon loss?

This is a great point that we simply hadn't considered before. In retrospect, we do not have a logical response; we were simply in the mindset of conducting field recordings for virally-injected mice due to prior use of TeTX for silencing, which could not have been tested by whole-cell recordings. We chose 7 dpi for slice recordings in virally-injected animals as a time point that balanced the known delay between injection and viral expression against the goal of measuring transmission before axon loss occurred. We settled on 7 dpi based on histological data from TeTX and Kir animals showing robust YFP expression in the DG at that time point, followed by the first signs of axon disintegration in the TeTX mice shortly after. The reviewer is correct that we cannot rule out the possibility that initial axon loss contributed to the observed transmission deficit, but we believe the time point we chose was the best compromise for the experiment.

– I might have missed it in the main text, but I am wondering if the observation about the absence of axon retraction bulbs in the legend of Suppl. 2 should not be in the main text. I finished my first reading of the paper with the impression that only the Tetanus toxin experiment was not showing the same retraction bulbs.

We thank the reviewer for this point and agree that this observation should be clearly stated in the main text. We have revised the Results section *EC2 cell death is observed with other means of electrical inactivation* based on this recommendation:

“Unlike the GlyCl model, we did not see retraction bulbs in the EC of Kir2.1 mice, but did observe evidence of axon damage in the DG at later time points.”

– It would be great to show the c-fos stainings post TTX infusion and seizure induction. Was c-fos staining eliminated in the DG or in ECII or both? In the LECII?

We agree that this is a great suggestion and have now added c-fos staining to validate our TTX infusions. This data is now provided as Figure 6 – Supplementary Figure 1.

– When thinking of the differences between the TeTx and GlyCl, I was wondering if you could maybe give a sense of the proportion of mECII cells that were targeted by the stereotaxic injection? Is it the entirety of mECII that is transduced? Do you see any difference in the density/thickness of the MML between the GlyCl YFP and the TeTx-YFP before and after degeneration and at different dorso-ventral levels? I do not think that this is essential data, but since you are making the comparison between inhibition of degeneration by TTX in both paradigms, these might represent at least a discussion point that could explain some discrepancies. i.e. more or less competition in one case vs. the other?

We thank the reviewer for raising this point and agree that potential differences in the proportion of EC cells expressing GlyCl vs TeTX is an important caveat when interpreting the results of these two experiments. Due to the nature of stereotaxic viral targeting in a transgenic driver line (for TeTX) compared with using a germline transgenic responder (for GlyCl), we know that the proportion of EC cells silenced by TeTX was generally smaller than for GlyCl. The expression of TeTX was also more variable between animals than for GlyCl, and along the dorso-ventral axis within individual mice, for the same reason. We cannot rule out the possibility that these differences affected the degenerative phenotype observed in each system. However, in all cases of TeTX delivery, regardless of EC transduction density or dorso-ventral location, we saw disintegration without retraction bulbs. This suggests that the degenerative phenotype was a result of the silencing mechanism and not the efficacy of viral delivery. With regard to the TTX experiments, we again agree that differences in the proportion of EC cells silenced by GlyCl vs TeTX could well have contributed to the rescue from degeneration observed in one case and not the other. However, TeTX expressing animals responded to TTX regardless of transduction efficiency, which suggests to us that the difference from GlyCl is intrinsic to the silencing system. Nevertheless, this is an important point to consider and as suggested, we have added this discussion to the Discussion section *Limitations of our experimental models.*

“In addition, stereotaxic viral injections likely reached a smaller fraction of EC neurons than transgenic expression of GlyCl, which would alter the proportion of active vs silenced cells in each system. This could influence the level of competition within each system, which could alter the degenerative phenotype we observed. Nevertheless, TeTX expression always produced axon disintegration rather than retraction, regardless of transduction efficiency. We recognize that the proportion of silenced cells might also have affected the potential for rescue by broad TTX silencing. Similarly, axonal disintegration in TeTX mice was consistently rescued by TTX, independent of the number of cells transduced. Because of this, we conclude that the differences in degenerative phenotypes between GlyCl and TeTX are most likely intrinsic to the silencing system, but appreciate the need for additional studies using matched viral or transgenic expression systems.”

– In the part "Preferential resilience of Wfs1 cells within EC2" in the discussion, I am not sure I understand "but the changeable pattern of cell loss we found between neighboring regions and within EC2 suggests that molecular signatures must be just one part of a complex balance": do you imply that these neighboring cells are pretty close in terms of molecular signatures, and so that molecular signatures are not the only determinant of vulnerability? Pyramidal and stellate cells of mECII do have a number of molecular differences that could account for their differential vulnerability including Wfs1 expression as you mention. But maybe I misunderstood the sentence.

We agree that this sentence is not as clear on paper as it was in our heads. We had intended to convey that two important distinctions between the two cell types are their connectivity and function within the hippocampal circuit, but do not mean to argue that the molecular signatures of stellate and pyramidal cells are similar or irrelevant in their differential vulnerability. We have modified the Discussion section *Preferential resilience of Wfs1 cells within EC2* in an effort to make this point more clearly:

“The factors contributing to differential vulnerability between stellate and pyramidal neurons are only just being elucidated, but the distinct patterns of cell loss we found within EC2 suggest that molecular signatures may be just one piece of this puzzle. Our data suggest that connectivity and function within the hippocampal-EC circuit may be equally important in governing differential neuronal vulnerability.”

– In the last section of the discussion, the authors say "For example, formation of neurofibrillary tangles in EC neurons might impair…". I am wondering if the authors should not replace NFT with a broader "tau pathology" or "pathological tau accumulation", considering the fact that there starts to be considerable evidence that neurofibrillary tangles are not necessarily the toxic tau species dysregulating neuronal excitability.

Well said. We have adjusted this point in the Discussion section *Vulnerability of the EC-hippocampal circuit may derive from its function* as suggested.

“For example, pathological tau accumulation in EC neurons might impair the initiation or propagation of action potentials in affected cells while neighboring neurons remain normally active.”